# The Role of Plant-Derived Natural Products as a Regulator of the Tyrosine Kinase Pathway in the Management of Lung Cancer

**DOI:** 10.3390/cimb47070498

**Published:** 2025-06-30

**Authors:** Faris Alrumaihi, Arshad Husain Rahmani, Sitrarasu Vijaya Prabhu, Vikalp Kumar, Shehwaz Anwar

**Affiliations:** 1Department of Medical Laboratories, College of Applied Medical Sciences, Qassim University, Buraydah 51452, Saudi Arabia; f_alrumaihi@qu.edu.sa (F.A.);; 2Department of Biotechnology, Microbiology and Bioinformatics, National College (Autonomous), Tiruchirapalli 620001, India; 3Department of Medical Laboratory Technology, Mohan College of Nursing and Paramedical Sciences, Bareilly 243302, India

**Keywords:** natural product, lung cancer, pathogenesis, tyrosine kinase, tyrosine kinase inhibitors, tyrosine-kinase-inhibitor-acquired resistance

## Abstract

One of the most common malignant tumors worldwide is lung cancer, and it is associated with the highest death rate among all cancers. Traditional treatment options for lung cancer include radiation, chemotherapy, targeted therapy, and surgical resection. However, the survival rate is low, and the outlook is still dreadfully dire. The pursuit of a paradigm change in treatment approaches is, therefore, imperative. Tyrosine kinases (TKs), a subclass of protein kinases, regulate vital cellular function by phosphorylating tyrosine residues in proteins. Mutations, overexpression, and autocrine paracrine stimulation can transform TKs into oncogenic drivers, causing cancer pathogenesis. Tyrosine kinase inhibitors (TKIs) have emerged as an attractive targeted therapy option, especially for non-small cell lung cancer (NSCLC). However, resistance to TKIs, and adverse cardiovascular effects such as heart failure, atrial fibrillation, hypertension, and sudden death, are among the most common adverse effects of TKIs. There is increasing interest in plant-derived natural products in the hunt for powerful chemosensitizer and pathway modulators for enhancing TKI activity and/or overcoming resistance mechanisms. This highlights the mechanism of TKs’ activation in cancer, the role of TKIs in NSCLC mechanisms, and the challenges posed by TKI-acquired resistance. Additionally, we explored various plant-derived natural products’ bioactive compounds with the chemosensitizer and pathway-modulating potential with TKs’ inhibitory and anticancer effects. Our review suggests that a combination of natural products with TKIs may provide a novel and promising strategy for overcoming resistance in lung cancer. In future, further preclinical and clinical studies are advised.

## 1. Introduction

Plants are essential sources of medications for treating a variety of diseases, and, since ancient times, people have used treatments based on medicinal plants to treat a wide range of diseases [1]. Numerous traditional and contemporary medical systems make use of various plant parts, such as stems, bark, leaves, flowers, seeds, and many more. Various plant extracts have been employed as fitness promoters in Saudi Arabia, India, China, the United States, Germany, and France in recent years. Out of the 300,000 plant species, only 15% have been processed to regulate their pharmacological characteristics, as was anticipated. Therefore, it is more encouraged to produce innovative medications using natural resources. Plants include a variety of vital compounds, including essential amino acids, lipids, vitamins, minerals, carbs, fiber, flavonoids, and other phenolic compounds [2]. Natural sources provide the model for modern synthesized drugs and drug-like chemicals. Numerous studies have been conducted on immunomodulators derived from natural sources to prevent diseases and modify immune function [3,4].

Cancer is a multifaceted disease characterized by the uncontrolled growth and spread of abnormal cells, which can invade surrounding tissues and metastasize to distant organs [5]. It remains a significant global health challenge, accounting for approximately 10 million deaths annually, making it one of the leading causes of mortality worldwide. Advances in understanding the molecular mechanisms underlying cancer have revealed its complexity, driven by genetic, epigenetic, and environmental factors. Recent research highlights the intricate interplay between cancer cells and the tumor microenvironment, including immune cells, stromal components, and extracellular matrix, which collectively influence tumor progression and response to therapy [6].

The formation of reactive oxygen and nitrogen species, which can directly cause DNA damage in epithelial cells, is a significant mechanism by which chronically activated neutrophils and macrophages directly contribute to the malignant conversion of epithelial cells [7,8]. DNA damage generated by external stimuli is a major element in the transformation of normal cells into cancer cells, and the cell response to DNA damage is an early event aimed at preventing carcinogenesis [9]. Therefore, DNA damage repair activity has a dual purpose in malignancies. Previous research has demonstrated that boosting DNA repair activity can suppress carcinogenesis or enhance apoptosis. However, it is unclear whether DNA repair activity plays a significant role in the promotion of cancer cells, and how medications with higher DNA repair activity affect cancer cells [10]. A feature of cancerous cells in culture is their reduced need for serum or other growth stimuli [11].

Current trends in cancer research emphasize early detection, personalized medicine, and novel therapeutic approaches. The development of liquid biopsies, which analyze circulating tumor DNA and other biomarkers, has revolutionized cancer diagnostics by enabling the non-invasive and real-time monitoring of disease progression. Additionally, targeted therapies, including immune checkpoint inhibitors and small-molecule inhibitors, have transformed treatment paradigms for several cancer types, offering improved outcomes for patients. The integration of artificial intelligence and machine learning into cancer genomics and drug discovery is further accelerating the development of precision oncology. These advancements, along with a deeper understanding of cancer biology, offer hope for better prevention, diagnosis, and treatment strategies in the fight against cancer [12].

Lung cancer is the greatest cause of cancer-related mortality worldwide, accounting for over 25% of all cancer deaths. Despite recent advancements in diagnosis and treatment, the five-year survival rate remains low. Lung cancer is widely classified into two types [13]. One of the key signaling enzymes in cell signal transduction is protein tyrosine kinase (TK), which catalyzes the transfer of ATP-γ-phosphate to the substrate protein’s tyrosine residues, phosphorylating it and controlling a number of physiological and biochemical processes, including cell growth, differentiation, and death. TK expression abnormalities are closely linked to tumor invasion, metastasis, and angiogenesis and typically result in disorders of cell proliferation. Many TKs are currently being used as targets in antitumor drug screening. Tyrosine kinase inhibitors (TKIs) decrease tyrosine kinase phosphorylation and compete with ATP for the ATP binding site of TK, which stops the growth of cancer cells [14].

The development, spread, invasion, and metastasis of cancers are all linked to aberrant TK activation brought on by mutations, translocations, or amplifications. Furthermore, in cancer, wild-type TKs may also serve as essential nodes for pathway activation. As such, tyrosine kinases have emerged as key targets for drug development. The purpose of a tyrosine kinase inhibitor (TKI) is to prevent the corresponding kinase from catalyzing phosphorylation [15].

One promising treatment strategy for non-small cell lung cancer (NSCLC) is targeted therapy targeting the epidermal growth factor receptor (EGFR). However, one of the biggest obstacles to its clinical management is still its resistance to EGFR-tyrosine kinase inhibitors (EGFR-TKIs) [16]. TKIs have changed the way that many solid cancers are treated and significantly increased patient survival and quality of life. In this paper, we summarized the roles of TKs in cancer, TKI treatment pathways, and TKI-acquired resistance mechanisms. We also go over a number of natural compounds that may interfere with TKs and how they work in tandem with chemotherapy to treat cancer.

## 2. Molecular Pathogenesis of Lung Cancer

Lung cancer remains the leading cause of cancer-related deaths globally [17,18], with a notably poor prognosis despite advancements in diagnostic tools and treatment options [17]. Lung cancer metastasis is the process by which the initial malignant tumor in the lung leaves the main site and spreads at a distance in a variety of ways. Metastasis occurs most commonly in the brain, bone, lymph nodes, and liver. Lung cancer metastasis is a highly complex process that involves the lung cancer microenvironment and lung cancer stem cells (LCSCs) as well as a variety of processes such as EMT development, angiogenesis, and lymphangiogenesis. These influencing factors are related to several non-coding RNAs (ncRNAs), associated factors of lung cancer metastasis, and multiple signaling pathways [19]. The disease is primarily categorized into two main types: non-small cell lung cancer (NSCLC), which accounts for approximately 85% of cases, and small cell lung cancer (SCLC), comprising the remaining 15%. The predominant risk factor for lung cancer is smoking; however, other factors such as exposure to environmental pollutants, including fine particulate matter, also contribute to its development [17,18].

Lung cancer develops due to a combination of genetic mutations and environmental factors, with smoking being the most significant cause, responsible for over 80% of cases. Cigarette smoke contains carcinogens that damage DNA and induce mutations in key oncogenes and tumor suppressor genes. Additionally, exposure to secondhand smoke, air pollution, and occupational carcinogens such as asbestos, arsenic, and radon gas significantly increases the risk of lung cancer. Genetic predisposition and familial history also play a role in susceptibility. Non-smokers may develop lung cancer due to environmental factors, chronic inflammation, or genetic alterations, including EGFR mutations and ALK rearrangements [17].

Lung cancer is broadly categorized into two main types, non-small cell lung cancer (NSCLC) and small cell lung cancer (SCLC), based on the cellular and molecular characteristics of the tumor (Figure 1). These classifications are essential for determining appropriate treatment strategies and understanding disease prognosis.

### 2.1. Non-Small Cell Lung Cancer (NSCLC)

NSCLC carcinogenesis is a complex process comprising various causes, stages, and processes, similar to that of other malignant tumors. Avoidable and unavoidable risk factors are two further categories into which the etiology of NSCLC can be divided. Inhaled tobacco use is the most well-known preventable risk factor for non-small cell lung cancer. Alcohol consumption, secondhand smoke exposure, asbestos, radon, arsenic, chromium, nickel, ionizing radiation exposure, and polycyclic aromatic hydrocarbons are other causes of lung cancer. The tumor, node, metastasis (TNM) stage, the patient’s performance status, and any comorbidities all affect the prognosis of non-small cell lung cancer. Patients with subpar performance status have a lower chance of surviving. Weight loss and reduced appetite are other indicators of poor prognosis [18].

It has been demonstrated that squamous cell carcinoma (SCC), a subtype of NSCLC, has a stronger association with chronic obstructive pulmonary disease (COPD) (e.g., risk, lower overall survival, severity), as well as an increased risk of SCLC with the presence of COPD independent of smoking status. However, no biological relationship has been found [13].

NSCLC is the most common kind of lung cancer, accounting for 85% of all cases. NSCLC consists of lung adenocarcinoma, lung squamous cell carcinoma (LSCC), and lung giant cell carcinoma. Adenocarcinoma develops in the distal airway and is not associated with smoking. LSCC develops in the proximal airway and is more aggressive and strongly linked to smoking than adenocarcinoma. Large cell carcinoma develops in the distal airway, and the cancer cell mass is greater than in the other two forms of NSCLC. Large cell carcinoma is an aggressive tumor [20].

#### 2.1.1. Adenocarcinoma

Adenocarcinoma is the most common form of NSCLC and is frequently diagnosed in non-smokers and younger individuals. It typically originates in the outer regions of the lung and arises from glandular cells that secrete mucus. Adenocarcinoma is often associated with genetic mutations such as EGFR (epidermal growth factor receptor), ALK (anaplastic lymphoma kinase), and KRAS (Kirsten rat sarcoma viral oncogene homolog), making it a target for precision therapies [21].

#### 2.1.2. Squamous Cell Carcinoma (SCC)

Usually linked to smoking, this subtype develops in the central regions of the lungs, often near the bronchial tubes. It arises from the epithelial cells lining the airways and is characterized by a slower growth rate compared to other subtypes [22]. Squamous cell lung cancers frequently develop in the main airway, such as the left or right bronchus, or in the center of the lung. Smoking is the primary cause of cellular change. Smoking is linked to about 90% of lung cancer patients in women and 80% of cases in men. Smoking has a stronger correlation with SCC than any other kind of NSCLC. Age, family history, secondhand smoking exposure, and occupational exposure to minerals, metal particles, or asbestos are additional risk factors for lung squamous cell carcinoma [23].

#### 2.1.3. Large Cell Carcinoma

This is the least common subtype of NSCLC and is known for its aggressive nature and rapid growth. It can appear in any part of the lung and often presents diagnostic challenges due to its lack of distinct features. The diagnosis of LCLC is primarily based on post-operative pathological examination rather than biopsy and cytological examination. It is known to be an undifferentiated non-small cell carcinoma because it lacks the cellular and structural characteristics associated with adenocarcinoma or squamous cell carcinoma. Additionally, there is evidence that LCLC is more common in older adults (>60), that it is often found in males and smokers, and that it frequently presents as a large mass with central necrosis [24].

### 2.2. Small Cell Lung Cancer (SCLC)

SCLC represents about 15% of lung cancer cases and is strongly associated with heavy smoking. It is characterized by small, round, or oval-shaped cells that grow rapidly and form large tumors. SCLC is highly aggressive, with a tendency to metastasize early to other organs, such as the brain, liver, and bones. Due to its rapid progression, SCLC is typically staged as limited-stage or extensive-stage disease, which guides treatment strategies [25]. Small cell lung cancer (SCLC) accounts for 13% of all incidences and is strongly linked to tobacco use. The specific molecular pathways underlying SCLC pathogenesis are currently unknown. However, significant genetic and molecular changes have been observed in SCLC, including the development of autocrine growth loops, proto-oncogene activation, and the deletion or inactivation of tumor suppressor genes [26].

Approximately 70% of patients with SCLC had distant metastases at the time of first diagnosis. At the same time, SCLC spread to a variety of metastatic locations, including the lungs, liver, brain, bone, adrenal glands, and lymph nodes. SCLC is a kind of extremely aggressive and invasive neuroendocrine carcinoma with poor differentiation and prognosis. The histocytological study of clinical samples obtained through needle biopsy and bronchoscopy is a regularly used procedure in clinical diagnosis. Tumor metastasis occurs in four stages: invasion, penetration into the circulatory system, crossing the brain barrier, and brain colonization. During tumor invasion, tumor cells penetrate the basement membrane and transform from an in situ to an invasive tumor. During the epithelial–mesenchymal transition (EMT) process, epithelial cells lose their adhesion capacity and gain the ability of mesenchymal cells to invade, metastasize, and resist drugs [27].

Autocrine growth factors, such as neuroendocrine-regulating peptides (for example, bombesin/gastrin-releasing peptide), are prevalent in SCLC. The Myc family’s dominant oncogenes are routinely overexpressed in both SCLC and NSCLC, although the K-RAS oncogene is never mutated in SCLC but is in 30% of NSCLC. The most common genetic disorders include tumor suppressor genes (TSGs). The TSG p53 is mutated in more than 90% of SCLCs and more than 50% of NSCLCs; the retinoblastoma TSG is inactivated in over 90% of SCLCs but only 15% of NSCLCs; and p16, the other component of the retinoblastoma/p16 pathway, is almost never abnormal in SCLC but inactivated in more than 50% of NSCLCs. The FHIT TSG is inactivated in 50–70% of lung cancer [28].

### 2.3. Other Rare Subtypes

In addition to NSCLC and SCLC, there are rare subtypes of lung cancer, including carcinoid tumors, which are slow-growing neuroendocrine tumors, and pleural mesothelioma, which arises from the lining of the lungs and is commonly linked to asbestos exposure [29].

## 3. Treatment Strategies for Lung Cancer

Recent advancements in treatment strategies have shown promise in improving patient outcomes. Understanding these types is critical for optimizing treatment approaches, including surgery, chemotherapy, targeted therapy, and immunotherapy. The treatment of lung cancer by these approaches, either solely or in a combination of these approaches, depends on the stage and type of lung cancer. Research continues to explore molecular subtypes and biomarkers for better classification and personalized treatment options [29]. In NSCLC, the integration of immunotherapies, particularly immune checkpoint inhibitors targeting the PD-1/PD-L1 pathway, has led to significant and sustained responses in patients without targetable oncogenic driver alterations. Additionally, the exploration of therapeutic cancer vaccines combined with immune checkpoint inhibitors is underway to enhance antitumor immunity. For SCLC, ongoing research into its molecular characteristics and immunologic microenvironment aims to identify novel therapeutic targets, potentially leading to more effective treatments in the future [30,31].

## 4. EGFR Signaling: Molecular Insights and Clinical Advances

Numerous growth factors and cytokines that alter the course and behavior of cancer cells are also secreted by tumor-associated myeloid cells. These include transforming growth factor β (TGF-β), Interleukin-6 (IL-6), and Interleukin-10 (IL10) and glycoprotein nonmetastatic melanoma protein B (GPNMB), hepatocyte growth factor (HGF), and epidermal growth factor (EGF) [32,33]. As a common mitogenic factor, epidermal growth factor (EGF) promotes the growth of various cell types, particularly fibroblasts and epithelial cells, by activating the EGF receptor (EGFR/ErbB), which, in turn, triggers intracellular signaling. EGF promotes the growth of numerous cell types in tissue culture, such as keratinocytes and transformed cells. A fraction of NSCLC was identified to have active mutations of the epidermal growth factor receptor (EGFR) gene, and tumors with EGFR mutations were extremely responsive to EGFR-TKI [34].

By attaching itself to its membrane receptor, i.e., EGFR, EGF produces mitogenic effects. This receptor can then control cell growth, proliferation, differentiation, and survival, among other biological processes. Since EGFRs are expressed in a range of human tissues, such as fibroblasts, endothelial cells, and the majority of epithelial tissues, EGF is essential for wound healing and tissue integrity [35].

EGFR plays a pivotal role in the pathogenesis of lung cancer, particularly in NSCLC. EGFR is a transmembrane tyrosine kinase receptor that regulates cell proliferation, survival, and differentiation through downstream signaling pathways such as phosphatidylinositol 3-kinase/protein kinase B (PI3K-AKT), mitogen-activated protein kinase (MAPK), and the Janus kinase/Signal transducer and activator of transcription (JAK-STAT). Mutations in the EGFR gene, commonly found in exons 19 and 21, are prevalent in a subset of NSCLC patients, especially in non-smokers and individuals of Asian descent. These activating mutations result in constitutive receptor signaling, driving oncogenesis and tumor progression [36].

Targeting EGFR mutations has revolutionized lung cancer treatment through the development of EGFR tyrosine kinase inhibitors (TKIs). First- and second-generation TKIs, such as erlotinib, gefitinib, and afatinib, have demonstrated significant efficacy in patients with EGFR-mutant NSCLC. However, resistance often emerges due to secondary mutations, such as T790M in exon 20. Third-generation TKIs, like osimertinib, have been developed to overcome these resistance mechanisms and have become the standard of care for patients with advanced EGFR-mutant NSCLC. Recent studies have also explored the role of EGFR in the tumor microenvironment and immune evasion, opening avenues for combination therapies with immune checkpoint inhibitors. Advances in liquid biopsy technologies further enhance the detection of EGFR mutations, enabling the real-time monitoring of treatment response and resistance development [37,38].

## 5. Targeting Tyrosine Kinases (TKs) in Cancer Therapy: Molecular Mechanisms and Drug Development Strategies

Numerous physiological responses, including growth, differentiation, proliferation, survival, apoptosis, migration, and cell cycle regulation, are drastically altered when kinase activity is deregulated [39]. TKs are a type of protein kinase that is important for hematopoiesis. They include discoidin domain receptor (DDRs), erythropoietin-producing human hepatocellular (Eph) receptors, proto-oncogene c-Src (SRC), spleen tyrosine kinase (SYK), Fms-like tyrosine kinase 3 (FLT3), Janus kinase (JAK), and others. The development of hematological malignancies has long been linked to the disruption of TK signaling [40]. TKs make up a significant percentage of all oncoproteins and are involved in the transformation of numerous malignancies. As a result, finding and creating therapeutic agents for conditions associated with aberrant TK activation brought on by increased expression, mutation, or autocrine stimulation that results in aberrant downstream oncogenic signaling has become a key focus for cancer treatment [41].

The development, spread, invasion, and metastasis of cancers are all linked to aberrant TK activation brought on by mutations, translocations, or amplifications. Furthermore, in cancer, wild-type tyrosine kinases may also serve as essential nodes for pathway activation. As such, tyrosine kinases have emerged as key targets for drug development. The purpose of a tyrosine kinase inhibitor (TKI) is to prevent the corresponding kinase from catalyzing phosphorylation [15].

TKs selectively phosphorylate tyrosine residues in various substrates. When ligands attach to the extracellular domain of receptor tyrosine kinases, they become active. Extracellular signaling chemicals called ligands, such as PDGF, EGF, and others, cause receptor dimerization, with the exception of the insulin receptor. The methods used by various ligands to attain the stable dimeric conformation vary. In certain conditions, two ligands bind simultaneously to two receptors (2:2 ligand–receptor complex), which offers the most straightforward mechanism of receptor dimerization (e.g., VEGF and VEGFR). In other situations, one ligand may bind with two receptor molecules to form a 1:2 ligand–receptor complex, such as growth hormone and growth hormone receptor [41].

There are 90 distinct TK genes in the human genome; 58 of them are receptor types, which are divided into 20 subfamilies, and 32 are non-receptor types, which are divided into 10 subfamilies [40]. The two main categories of tyrosine kinases are non-receptor tyrosine kinase (NRTK), which includes SRC, ABL, FAK, and Janus kinase, and receptor tyrosine kinase (RTK), which includes EGFR, PDGFR, FGFR, and the IR. In addition to being transmembrane receptors on the cell surface, the receptor tyrosine kinases are also enzymes with kinase activity. The receptor tyrosine kinase’s structural arrangement includes a cytoplasmic section with a TK domain, a single-pass transmembrane hydrophobic helix, and a multidomain extracellular ligand for expressing ligand specificity. Both the N and C terminal ends of the kinase domain contain regulatory sequences [41,42,43].

Understanding the role and importance of TKs in cancer was made possible by the discovery that the SRC oncogene possesses changing non-receptor tyrosine kinase (NRTK) activity and that protein–tyrosine kinase (PTK) activities are associated with viral transforming proteins [41,44]. Targeting TKs represents an important approach of oncology. Advances in molecular biology, structural bioinformatics, and next-generation sequencing can open new realms of rational drug designing [14].

## 6. TKs in Lung Cancer: Molecular Pathways and Clinical Implications

Phosphatases are enzymes that are implicated in the removal of a phosphate group from a protein. However, kinases are enzymes that transfer a phosphate group to a target molecule and cause the phosphorylation of proteins. By controlling many signaling pathways in response to external stimuli, these two enzymatic activities are essential to the cell [40]. One of the pivotal molecular mechanisms underlying NSCLC progression is the dysregulation of TK signaling pathways. In NSCLC, the aberrant activation of receptor and non-receptor TKs contributes to tumorigenesis, metastasis, and resistance to conventional therapies [36]. TKs implicated in NSCLC can be classified into two broad categories: receptor tyrosine kinases (RTKs) and non-receptor tyrosine kinases (NRTKs). RTKs, such as EGFR, anaplastic lymphoma kinase (ALK), and fibroblast growth factor receptor (FGFR), are often mutated or overexpressed in NSCLC [41].

NSCLC carcinogenesis is a complex process comprising various causes, stages, and processes, similar to that of other malignant tumors. Its primary pathomechanism is the activation of a proto-oncogene or the inactivation of an anti-oncogene in response to a variety of circumstances, including low levels or poor DNA repair ability [45]. In human malignancies, four major mechanisms contribute to constitutive RTK activation: gain-of-function mutations, genomic amplification, chromosomal rearrangements, and/or autocrine activation. When ligands are present, intermolecular dimerization happens, activating RTKs by phosphorylating the receptor’s C-terminal tail and activating kinases. Usually without a ligand, the mutations cause the constitutive activation of the RTK. Increased local receptor concentration is caused by the overexpression of RTKs, which frequently occurs as a result of the genomic amplification of the RTK gene [46].

Chromosomal rearrangements culminate in the creation of a hybrid fusion oncoprotein made up of the RTK and the fusion partner, a separate protein. RTKs are often activated by receptor-specific ligands. Growth factor ligands bind to extracellular areas of RTKs, activating the receptor via ligand-induced receptor dimerization and/or oligomerization [47]. Most RTKs’ structural alterations allow for the trans-autophosphorylation of each TKD and the release of cis-autoinhibition [48]. This conformational shift enables the TKD to adopt an active configuration. RTK autophosphorylation recruits and activates a diverse set of downstream signaling proteins with Src homology-2 (SH2) or phosphotyrosine-binding (PTB) domains. These domains bind to certain phosphotyrosine residues within the receptor and engage downstream mediators, propagating crucial physiological signaling pathways [49].

RTKs are activated by intermolecular dimerization in the presence of ligands, which causes kinase activation and the phosphorylation of the receptor’s C-terminal tail. The alterations cause the RTK to be activated constitutively, even when no ligand is present. The overexpression of RTKs, sometimes caused by the chromosomal amplification of the RTK gene, leads to an increase in the local concentration of receptors. Depending on where the chromosomal breakpoint is, these RTK fusion proteins might be either membrane-bound or cytoplasmic. In both cases, the kinase domain becomes activated. In the absence of ligands, the duplication of the tyrosine kinase domain may result in the formation of an intramolecular dimer, which would activate RTK. The enhanced local ligand concentration activates the RTK, leading to RTK dimerization, enhanced kinase activity, and the phosphorylation of the receptor’s C-terminal tail [46].

EGFR mutations, particularly in exons 19 and 21, are associated with oncogenic signaling and poor prognosis. These mutations lead to the constant activation of downstream pathways like RAS-RAF-MEK-ERK, which promote uncontrolled cell proliferation and survival. Similarly, ALK gene rearrangements and FGFR amplifications also contribute to tumor growth and invasion through the activation of PI3K-AKT-mTOR and MAPK pathways [41,50]. Many reports have shown that the interplay between cancer-related pathways and TK signaling pathways contributes to tumor progression and complicates treatment due to resistance mechanisms.

In addition to RTKs, non-receptor tyrosine kinases, such as Src family kinases and Janus kinases (JAKs), play a significant role in NSCLC. The aberrant activity of these kinases is implicated in enhancing cell motility, invasion, and resistance to apoptosis. Src family kinases, in particular, are known to promote EMT and increase metastatic potential, making them attractive targets for therapeutic interventions [51]. Beyond their involvement in intracellular signaling pathways, TKs significantly influence the tumor microenvironment by modulating angiogenesis, immune cell infiltration, and inflammatory responses. In lung cancer, VEFGR-mediated signaling induces tumor vascularization and metastasis. Additionally, TKs have been reported for their ability to upregulate immune checkpoint molecules, suppress antitumor activity, and promote immune evasion, which contributes to the resistance towards treatment. Advanced techniques including liquid biopsy and NGS have been known for their real-time detection of TK alteration, leading to improvement in patient stratification and treatment monitoring and the early identification of resistance [41,46,50].

## 7. Advances in Targeted Therapy for Lung Cancer: Tyrosine Kinase Inhibitors (TKIs) and Resistance Pathways

Malignancy can result from the mutations, overexpression, and autocrine paracrine stimulation of TKs, even if their activity is strictly controlled in healthy cells. Because selective TKIs can prevent constitutive oncogenic activation in cancer cells, they are seen as a promising strategy for novel genome-based therapies. The recognition of TK dysregulation as a key driver in NSCLC has led to the development of tyrosine kinase inhibitors (TKIs), which have significantly improved patient outcomes, particularly in cases with EGFR mutations. TKIs have been reported to be more successful in managing NSCLC, and its application in other types of lung cancer, such as SCLC, is more explored. However, progress in SCLC is limited. TKIs are classified into various categories on the basis of their generation (first, second, or third), mode of action (competitive, allosteric, or multi-targeted), and target (receptor or non-receptor). First-generation EGFR inhibitors like gefitinib and erlotinib, and second-generation inhibitors such as afatinib, target the kinase domain of EGFR, preventing its activation and downstream signaling. However, resistance to these therapies often develops due to secondary mutations, such as the T790M mutation, which limits the efficacy of these first-line agents [52].

To overcome this challenge, third-generation TKIs like osimertinib have been developed to specifically target EGFR mutations, including the T790M mutation. These inhibitors offer prolonged progression-free survival and are currently considered the standard of care for EGFR-mutant NSCLC. Similarly, ALK inhibitors, including crizotinib, alectinib, and brigatinib, have shown efficacy in patients with ALK-rearranged NSCLC, providing a significant survival benefit [53]. Figure 2 shows the classification of TKIs into first-, second-, and third-generation inhibitors (Figure 2).

Despite these advancements, resistance to TKIs remains a major hurdle. Mechanisms such as the acquisition of secondary mutations in the kinase domain, activation of bypass signaling pathways, and epithelial-to-mesenchymal transition contribute to treatment failure. To address this, combination therapies that target multiple signaling pathways or incorporate immune checkpoint inhibitors are being explored in clinical trials [54].

TKs are pivotal enzymes involved in cellular signaling processes that regulate cell growth, differentiation, survival, and metabolism. The dysregulation of TK activity is commonly associated with various cancers, where the aberrant activation of these kinases contributes to oncogenesis, tumor progression, and metastasis. Targeting TKs in malignant cells can disrupt the cell signaling pathways implicated in tumor growth [41]. A promising therapeutic approach for NSCLC involves targeted therapy targeting EGFR with precision therapies [16].

## 8. Mechanisms of Action and Molecular Targets of TKIs

Given their role in driving tumor progression, aberrant TKs have become central therapeutic targets. The purpose of a TKI is to prevent the corresponding kinase from catalyzing phosphorylation [15]. By targeting aberrant TKs, TKIs block phosphorylation, disrupt signaling, and induce apoptosis in malignant cells. Over the past few decades, TKIs have emerged as a cornerstone in the treatment of several malignancies, offering targeted therapeutic strategies with the potential for better specificity and fewer side effects compared to conventional chemotherapy [41]. TKIs can be bivalent, allosteric, or ATP-competitive [55].

TKIs inhibit TK activity by competitively binding to the ATP binding site. They can also influence activity by binding to other sites, which may block access to the Cdc37-Hsp90 chaperone system and, ultimately, inhibit downstream signaling pathways such as PI3K/Akt and Raf/MeK/Erk. TKIs have the ability to contend with ATP for tyrosine kinases’ ATP binding site. This stops signal transduction pathways from becoming phosphorylated and activated. TKIs can also attach to the human epidermal growth factor receptor (EGFR) kinase area. TKIs also exert their effects by binding to allosteric sites. Therefore, they disrupt the access to chaperone systems including Cdc37-HSP90 and, thereby, they interfere with kinase activity and function. In brief, TKs rely on the molecular chaperone system for cellular stability, but TKIs can prevent them from accessing it [14].

RTKs, such as EGFR, human epidermal growth factor receptor 2 (HER2), and VEGFR, are frequently mutated or overexpressed in various kinds of cancer, leading to uncontrolled signaling and tumor growth. In contrast, NRTKs, such as Src family kinases and Janus kinases (JAKs), are known to be implicated in cancer cell survival, migration, and invasion [56]. TKIs are designed to inhibit the kinase activity of these dysregulated TKs, preventing the phosphorylation of tyrosine residues on substrate proteins and, thus, halting the downstream signaling pathways that drive cancer cell proliferation. From this point of view, TKIs can be broadly categorized into small molecules and monoclonal antibodies. Small-molecule TKIs, such as imatinib, gefitinib, and erlotinib, typically target the intracellular kinase domains of RTKs, while monoclonal antibodies, such as trastuzumab and cetuximab, bind to the extracellular ligand-binding domains of RTKs, preventing receptor activation [57].

EGFR is a transmembrane glycoprotein which comprises an extracellular ligand-binding domain and an intracellular tyrosine kinase domain that modulates the main signaling pathways regulating cell growth and survival. Oncogenic mutations in EGFR are frequently associated with NSCLC and are responsible for aberrant cellular proliferation. When the EGFR binds to its ligand, intrinsic tyrosine/kinase activity causes autophosphorylation, which sets off a number of signal transduction cascades. The persistent activation of certain downstream target sequences is believed to contribute to more aggressive tumor behaviors. Mutations in epidermal growth factor receptor are commonly associated with various lung malignancies. Tyrosine kinase inhibitors significantly increase the responsiveness of lung adenocarcinomas with mutant epidermal growth factor receptors, but they do not seem to improve survival for unselected individuals [58].

## 9. Therapeutic Role of TKIs Across Cancer Types Including Lung Cancer

Several TKIs have demonstrated significant efficacy in clinical settings, particularly in cancers driven by specific mutations or the overexpression of TKs. Imatinib, the first TKI approved for clinical use, targets the BCR-ABL fusion protein in chronic myelogenous leukemia (CML), providing a revolutionary treatment option for CML patients with Philadelphia chromosome-positive tumors. Other TKIs, such as gefitinib and erlotinib, target EGFR mutations in NSCLC, leading to improved progression-free survival and overall response rates in patients harboring sensitive EGFR mutations [59].

In breast cancer, trastuzumab targets HER2-positive tumors, offering a significant survival benefit for patients with HER2 amplification. Other HER2-targeting TKIs, such as lapatinib, provide an option for patients who develop resistance to trastuzumab. Similarly, TKIs targeting vascular endothelial growth factor receptor (VEGFR), such as sorafenib and sunitinib, are used to treat renal cell carcinoma and hepatocellular carcinoma by inhibiting angiogenesis, a process essential for tumor growth and metastasis [60].

It is believed that more aggressive tumor behaviors result from the constitutive or persistent activation of certain downstream target sequences. Mutations in EGFR have been reported in relation to various lung malignancies. TKIs significantly increase the responsiveness of lung adenocarcinomas with mutant EGFR, but they do not seem to improve survival for unselected individuals [58].

Recent efforts have been made to evaluate aberrant kinases including FGFR1, AXL, and IGF-1R as potential therapeutic targets for the treatment of SCLC [43,45]. In NSCLC, multiple oncogenic drivers such as EGFR, AK, ROS1, MET, and BRAF have led to the approval of various TKIs with improved survival and disease control [44,46]. Targeted therapy against EGFR has been considered to be a promising treatment option for NSCLC, especially in patients with EGFR mutations. Moreover, their clinical benefits have a significant limitation on the eventual development of resistance. This resistance is a significant challenge [16]. Numerous studies have demonstrated that blocking TKs in cancerous cells with monoclonal antibodies, interfering RNAs, and/or tiny kinase inhibitors reduces cell survival and proliferation, causing growth arrest and apoptosis [61]. Clinical trials linked to various TKIs are provided in Table 1.

## 10. Challenges for Implicating TKIs in Lung Cancer: Sensitivity and Resistance

Drug resistance can be classified as intrinsic (primary) or acquired (secondary) resistance to antiangiogenic targeted therapies. Intrinsic resistance occurs when cancer cells are naturally insensitive to TKIs and do not respond to the drug, whereas acquired resistance occurs when cancer cells respond to TKI treatment initially but subsequently relapse as the drug loses efficacy over time due to the acquisition of various resistance mechanisms [67]. In aberrantly active EGFR, through mechanisms comparable to wild-type receptors, mutant EGFRs (caused by the deletion of exon 19 or punctual mutation in exon 21 known as L858R) exhibit higher levels and durations of EGFR activation. RAS/RAF/MEK/MAPK, phosphoinositide 3-kinase (PI3K)/AKT, and STAT3/STAT5 pathways can all be activated by mutated EGFR (Figure 3a) [68].

Because TKIs target important pathways involved in cancer proliferation, survival, and metastasis, they have greatly improved the prognosis for patients with advanced NSCL [69]. EGFR TKIs have demonstrated efficacy in managing EGFR-mutated lung cancer after several years of usage [68] (Figure 3b). On the other hand, resistance and subsequent disease progression are nearly always associated with VEGF-targeted TKI treatment. The immunosuppressive nature of the tumor microenvironment in lung cancer often prevents the efficacy of TKIs and also promotes resistance by interfering with signaling pathways. The various mechanisms that underlie TKI resistance, such as the upregulation of alternative proangiogenic pathways, EMT, efflux pumps that lower intracellular drug concentrations, lysosomal sequestration, changes in the tumor microenvironment, and genetic factors, make it difficult to mitigate drug resistance [67].

Lung cancer, particularly NSCLC, is characterized by a considerable frequency of heterogeneity, and this heterogeneity presents a significant obstacle to the uniform efficacy of TKIs. An excellent illustration is the well-known targeted treatment for NSCLC known as EGFR-TKI, which blocks overexpressed EGFR. The identification of EGFR-activating mutations marked a mile stone in the development of EGFR-TKIs, as these mutations were found to predict differential treatment response [70]. Regretfully, after a median of 9 to 14 months, resistance has been observed in EGFR-mutant patients on EGFR-TKI therapy [71,72]. In NSCLC, acquired resistance against first-generation EGFR-TKIs has been primarily linked to the acquisition of T790M mutation, which alters the kinase domain and inhibits drug-binding affinity [70]. The histological transformation from NSCLC to the SCLC phenotype has been reported to be an important mechanism of resistance, which makes therapeutic approaches more complicated.

Although TKIs show an effective initial response, their long-term potential is often limited due to the development of resistance, which is a significant challenge. Resistance to TKIs can occur through several mechanisms, including the development of secondary mutations in the kinase domain, activation of alternative signaling pathways, and upregulation of drug efflux pumps. For example, in NSCLC, resistance to first-generation EGFR TKIs is often associated with the acquisition of the T790M mutation, which alters the binding affinity of these inhibitors. In addition, intratumoral heterogeneity and clonal evaluation may promote the outgrowth of resistant subclonal populations characterized by additional oncogenic alterations beyond EGFR, such as MET amplification and HER2 mutations. To overcome resistance, third-generation EGFR TKIs, such as osimertinib, have been developed to specifically target the T790M mutation and exhibit improved clinical outcomes [73].

In addition, combination therapies that pair TKIs with other therapeutic modalities, such as immune checkpoint inhibitors or chemotherapy, are being actively explored to overcome resistance and improve treatment efficacy. Clinical trials are also investigating the potential of novel TKIs targeting less explored tyrosine kinases and the development of personalized medicine approaches to tailor TKI treatment based on specific molecular profiles [74]. Since resistance mechanisms in advanced lung cancer are dynamic and adaptive, ongoing molecular monitoring is necessary to modify therapeutic approaches in real time.

## 11. Expanding Therapeutic Landscapes of Kinases: Complexities and Challenges in Designing Novel TKIs

In addition to their established role in cancer, protein kinases have been implicated in a wide range of diseases, including as immunological disorders, skeletal and craniosynostosis disorders, hematological and vascular disorders, neurological disorders, multiorgan disorders, and endocrine and metabolic disorders. Many of these diseases are caused by mutations in members of the kinase gene family. As a result, TKIs now are recognized as valuable therapeutic agents outside of oncology. For example, tofacitinib is the first TKI approved for the treatment of inflammatory disorders [75].

Protein kinases have, thus, emerged as important pharmacological targets. Currently, there are approximately 250 kinase therapeutic candidates under clinical investigation, and 37 small TKIs have been approved for clinical use globally. A study examined the target spectrum of 243 clinically tested kinase drugs using chemical proteomics and provided a comprehensive data resource outlining the target landscape of 243 clinically tested KIs [76].

The landscape of TKIs is evolving rapidly, highlighting the need for continued research to discover new targets and develop TKIs capable of treating a broader range of solid tumors. However, there are several challenges such as enhanced structural diversity, improving kinase selectivity, and minimizing the off-toxicity profiles of existing kinase inhibitors [77].

Moreover, with increasing resistance mutations, there is a growing interest in the development of fourth-generation TKIs that exploit novel mechanisms of action. The successful establishment of such TKIs may require the exploration of unique binding modes, rigorous clinical validation, and the identification of appropriate patient populations [78]. Expanding the clinical applications of TKIs into non-oncology domains and establishing the effective combination approaches for cancer treatments require specifically designed TKIs. This undoubtedly presents a demand for more advanced research in medicinal chemistry, structural biology, and rational drug designing to ensure the discovery of safe, selective, and efficacious TKIs [75].

## 12. Plant-Derived Natural Products in Cancer Therapy: Modulating Apoptosis, Cell Signaling, and Chemo-Resistance

Numerous natural compounds with a wide range of structures are produced by plants. In contrast to the “primary metabolites,” which are necessary for the growth and development of plants, these products are often referred to as “secondary metabolites”; these natural compounds play crucial roles in how plants interact with their biotic and abiotic surroundings. For instance, they can function as hormones or signal molecules, floral colors that draw pollinators, or defensive substances against infections and herbivores. Natural products have a significant cultural influence in addition to their physiological role in plants. Throughout human history, they have been utilized as pharmaceuticals, condiments, and pigments [79].

As a rich source of therapeutically relevant biomolecules for the creation of new medications, natural products and secondary metabolites have already been mentioned. Obesity, diabetes, and cardiovascular disease are just a few of the chronic diseases that are greatly increased by poor eating. Chronic disease will most likely be exacerbated by food deficits resulting from an inadequate diet. Diets heavy in sodium and low in whole grains, fruit, vegetables, nuts, and seeds have been linked to higher death rates [80].

When compared to synthetic compounds, naturally occurring phytochemicals and products have been determined to be somewhat safe for human consumption. They are also reasonably non-toxic, affordable, and available in an ingestible form [81]. Over centuries, numerous plants and their products have been incorporated into traditional remedies, many of which have inspired modern pharmaceuticals [82]. It has previously been mentioned that plant-derived natural products and secondary metabolites are recognized as important sources of pharmacological compounds for drug development. New medications have been transformed by antibiotics (like penicillin, tetracycline, and erythromycin), antiparasitics (like avermectin), antimalarials (like quinine and artemisinin), lipid control agents (like lovastatin and analogs), immuno-suppressants for organ transplants (like cyclosporin and rapamycins), and anticancer medications (like doxorubicin and paclitaxel) [5].

A significant portion of our daily diet consists of fruits and vegetables, which are rich in polyphenolic chemicals with beneficial biological and pharmacological properties [83]. Because they have fewer adverse effects and can help reduce resistance to cancer treatment, medicinal plants or their bioactive constituents are dynamic sources of medications [84].

One of the leading causes of mortality worldwide is cancer. Current treatment methods, such as chemotherapy and radiation therapy, have a number of negative health impacts on individuals. According to this perspective, the bioactive component of natural products is essential for managing disease since it inhibits and activates biological processes like inflammation, oxidative stress, and cell signaling molecules. Natural products can be useful adjuvants or a kind of supporting therapy, but they are not a replacement for medications [85].

Because of their great effectiveness and minimal toxicity, natural products are gaining more and more attention. Numerous active components derived from herbs are frequently employed as antitumor medicines, which can increase anticancer efficacy and lessen adverse effects [86]. The US Food and Drug Administration (FDA) has licensed numerous medications derived from plants for use in cancer therapy, including taxanes like paclitaxel and vinca alkaloids like vinblastine. Dietary supplements are another type of natural product that is commonly used by cancer patients, but they lack the FDA-reviewed evidence of safety and efficacy—whether linked to survival, palliation, symptom reduction, and/or immune system enhancement—that is required for therapeutic approval. Nearly half of cancer patients in the United States report that they began using new dietary supplements after receiving their diagnosis. Oncologists face challenges when advising patients on which supplements are safe and effective to use to treat cancer or the adverse effects of cancer therapy [87].

In lung cancer specifically, several natural compounds have shown promise and potential antitumor activity with a favorable safety profile, while the combinatorial action of an anticancer drug with a natural compound provides synergistic action which helps boost the overall therapeutic action against cancer cells. In cancer, there is a dysregulation of apoptosis that facilitates the survival of the cancer cell, resulting in the progression of cancer. Many cancer drugs cause mutations of genes that regulate cancer and should kill the cancer cell but lead to chemoresistance. Many bioactive natural molecules modulate cancer-related cell signaling pathways, restore apoptosis, and exert cytotoxic effects in the target tumor cells. The importance of these compounds is emerging in many therapies developed with dual action, often including a natural compound [88].

## 13. Targeting TKI-Resistant Lung Cancer: Therapeutic Promise of Plant-Derived Natural Products

According to a recent review, 67 recognized natural compounds have the ability to fight cancer’s resistance to EGFR-TKIs through at least 30 pathways, primarily ROS, PD-L1, EGFR, MAPK, mTOR, HSP90, JNK, PTEN, and FOXO [86]. Additionally, various previous studies have documented the role of natural products such as TKIs against multiple kinds of cancers (Table 2). EGFR-TKI resistance is one of the biggest issues in cancer treatment, especially in NSCLC. Even though several studies, both in preclinical and clinical trials, have shown the promising therapeutic effects of polyphenolic compounds in anticancer therapy, the function of the natural compounds in TKI-resistant (TKIR) lung cancer remains poorly studied. Polyphenolic substances, including equol, kaempferol, resveratrol, ellagic acid, p-Coumaric acid, hesperidin, and gallic acid, were significantly reported to reduce cancer growth in TKIR cell H1993 while preserving TKIS cell H2073. When combined, our research offers a basic yet significant screening of possible natural chemicals for the development of anticancer drugs, particularly to overcome TKI resistance in advanced lung cancer. As a result, giving polyphenolic substances to patients with lung cancer may be a powerful adjuvant therapeutic approach that supports long-term TKI treatment [89].

The therapeutic benefits of EGFR-TKIs are enhanced by bioactive substances found in natural products such as alkaloids, saponins, terpenoids, polyphenols, resins, nucleosides, and quinones with significant tyrosine kinase inhibitory activity. By blocking the phosphorylation process and interfering with the signal transmission in the pathway, they can prevent the excessive activation of tyrosine kinase and cure cancer by competing with the ligand and ATP for binding sites on the kinase. Curcumin, resveratrol, ginsenosides, astragaloside IV, cucurbitacin D and cucurbitacin B, apigenin, quercetin, betulinic acid, β-elemene, licochalcone, sulforaphane, EGCG, and shikonin are among the natural chemicals that have significant glycolysis-inhibiting properties. These substances target a variety of glycolytic components and offer promising treatment options for EGFR-TKI resistance and cancer cell growth inhibition.

### 13.1. Alkaloids

#### 13.1.1. Capsaicin

Capsaicin is a bioactive alkaloid and is found in *Capiscum annum* L. Capsaicin exhibits significant antimetastatic effects in human fibrosarcoma (HT-1080 cells) by targeting EGFR-dependent signaling pathways. It inhibits the EGF-induced activation of MMP-2 and MMP-9, which are implicated in extracellular remodeling and tumor invasion [90]. Mechanistically, capsaicin suppresses EGFR-mediated downstream signaling cascades, including FAK/AkT, PKC/Raf/ERK, and p38 MAPK pathways, as well as AP-1 transcriptional activity, leading to a marked reduction in MMP-9 expression and tumor cell migration [91].

#### 13.1.2. Oxymatrine

Oxymatrine is derived from *Sophora flavescens* and it exhibits potent anticancer activities against various malignancies. In human malignant glioma (U251MG) cells, it was found to inhibit cell growth, induce the arrest of the cell cycle at the G0/G1 phase, and suppress the expression of key cell cycle regulatory proteins, thereby hindering tumor proliferation [92]. In gastric cancer cells, oxymatrine decreased the proliferation and invasion of gastric cells by inhibiting the EGFR/Cyclin D1/CDK4/6, EGFR/Akt, and MEK-1/ERK1/2/MMP2 pathways by inhibiting EGFRp-Tyr845. This results in reduced cancer cell proliferation and invasion, underscoring oxymatrine’s potential as an EGFR-targeted therapeutic agent [93].

#### 13.1.3. Tatrandrine

Tatrandrine is an alkaloid and is obtained from *Stephania tetrandra* S. Moore. Tatrandrine has demonstrated promising anticancer activities. In human colorectal adenocarcinoma (HT29), it effectively inhibits the phosphorylation of EGFR and its downstream signaling pathways, thereby suppressing tumor cell growth and survival mechanisms [94].

### 13.2. Flavonoids

#### 13.2.1. Apigenin

Apigenin, an active component of many Chinese medicinal herbs, has been widely studied for its anticancer properties and underlying mechanisms of action. Under in vivo conditions, apigenin is often conjugated to a glycoside in its native state, and apigenin was categorized as a class II medicine under the Biopharmaceutical Classification System [129]. Traditionally used in herbal medicines, apigenin has gained attention for its broad pharmacological activities. Among flavonoids, apigenin is recognized for its antioxidant, anti-proliferative, and carcinogenic effects. In addition to such benefits, the tumor-suppressing capability of apigenin has been documented in both historical and contemporary studies. Apigenin exhibits antitumor activity towards a variety of cancerous tumors using both in vitro cell lines and in vivo mice models [84].

Apigenin, which is technically known as 4′,5,7-trihydroxyflavone, is a member of the flavone family and is widely available in fruits, vegetables, and drinks. The foods rich in apigenin include celery, parsley, artichokes, and oregano, as well as beverages like red wine and beer. The dried flowers of *Matricaria recutita* L. are used to make chamomile tea, which is a rich source of apigenin. The chamomile flower head contains around 16.8% apigenin [130]. Because of its antioxidant and anti-inflammatory activities [131,132], its ability to lower blood pressure [133], and its antibacterial and antiviral properties [134], apigenin has been considered a multifunctional therapeutic agent.

Apigenin and cetuximab have been shown to inhibit the expression of p-STAT3, p-Akt, p-EGFR, and Cyclin D1 [135]. Apigenin and gefitinib’s combined effects on non-small cell lung cancer with a mutated epidermal growth factor receptor (EGFR) were assessed. Apigenin and gefitinib were found to inhibit several oncogenic drivers, including HIF-1α, EGFR, and c-Myc, and to decrease the expression of the MCT1 and Gluts proteins. As a result, treating lung cancer with apigenin and gefitinib together offers an appealing alternative therapeutic option for acquired resistance to epidermal growth factor receptor TKIs [136].

Apigenin can potentiate the antitumor effect of chemotherapeutic agents and/or alleviate the side effects of many anticancer agents. Due to the fact that TKIs are mostly metabolized by CYP3A4 enzymes and that apigenin could alter enzymatic activity, potential drug interactions could be expected following their co-administration [137]. Apigenin and kaempferol have been shown to inhibit EGFR, HER2, and MEK1, potentially contributing to the systemic prevention of metastatic colorectal cancer (mCRC). Furthermore, kaempferol and apigenin both exhibited dose-dependent antiangiogenic effects [138]. By specifically targeting RTKs, apigenin can influence lung cancer cell cycle progression, apoptosis, and EMT, highlighting the importance of flavonoid-mediated RTK inhibition [139,140]. Furthermore, it has been discovered that apigenin inhibits VEGF expression in human umbilical artery endothelial cells (HUA-EC) via HIF-1α. It was demonstrated that apigenin inhibits VEGF expression by degrading HIF-1α and interfering with Hsp90 activity [141]. Apigenin suppressed Akt and p70S6K1 activation, which may have been a consequence of VEGF suppression. Additionally, it has been demonstrated that apigenin in particular lowers the quantity of HIF bound to p300 and HIF-1α expression in lung cancer cells (NCI-H157) [142].

#### 13.2.2. Baicalein

The Src family of kinases includes the widely expressed non-receptor TKs (Src tyrosine kinases). Although Src family kinases are well known for their functions in the development of cancer, they also play a part in chemotaxis, proliferation, and signaling pathways linked to inflammation. Following the activation of multiple receptor types, Src is essential for attracting a variety of cell signaling molecules, which causes the production of several cytokines, including IL-6 [143]. There have been reports of baicalein’s many biological effects. In addition to exhibiting antiviral and antitumor actions, it is well known for its anti-inflammatory, antipyretic, and antihypersensitivity qualities. Human gastric, colon, hepatoma, pancreatic, and prostate cancer cells have all been shown to undergo apoptosis when exposed to baicalein. It has also been demonstrated to target metastasis and tumor angiogenesis [144].

Baicalein reduces the growth of tumor cells in non-small cell lung cancer (NSCLC) via inducing apoptosis, according to research by Leung and coworkers. This effect is associated with changes in the regulation of cell cycles and the altered expression of apoptotic regulatory proteins, including p53, caspase-3, and the bcl-2/bax ratio [97]. Because flavonoids may lower reactive oxygen species, which damage cells and tissues and raise the risk of inflammatory disorders, they are useful in the treatment of a variety of diseases. The cytotoxicity and anti-inflammatory properties of two flavonoids, baicalin and baicalein, which are present in the roots of *Scutellaria baicalensis* (*S. baicalensis)* and the leaves of *Thymus vulgaris* and *Oroxylum indicum*, were examined. Baicalein was shown to be the more potent substance, exhibiting greater inhibitory effects on Src tyrosine kinase and cytokine IL-6 production. Baicalein was shown to be the more potent substance, exhibiting greater inhibitory effects on Src tyrosine kinase and cytokine IL-6 production [145]. Additionally, baicalin and baicalein from *S. baicalensis* have shown possible inhibitory effects against EGFR tyrosine kinase activity in experiments [146].

#### 13.2.3. Curcumin

Diferuloylmethane, or curcumin, is extracted from *Curcuma longa*’s rhizome. Curcumin exhibits anti-NSCLC properties by modulating caspase-3 activity and miR-192-5p expression, leading to phosphoinositide 3-kinase (PI3K)/Akt signaling pathway inhibition and apoptosis [147]. By inhibiting the Wnt/β-catenin pathway, which is triggered by the metastasis-associated protein-1 (MTA-1), curcumin also inhibits the growth and invasion of NSCLC [148]. MTA-1 facilitates the invasion and metastasis of NSCLC cells [149]. The PI3K/Akt signaling pathway in NSCLC cells is suppressed and curcumin’s effects on cell viability and death are amplified by miR-192-5p mimics, but anti-miR-192-5p mimics have the opposite effect [147].

Additionally, curcumin therapy has been documented for its potential to suppress the development of NSCLC by lowering MTA-1. Curcumin induces autophagy in NSCLC, and this effect is reversed by autophagy inhibitor 3-methyladenine (3-MA). This effect exhibits the role of autophagy in anticancer activity [150]. Curcumin potentiates the therapeutic effects of gefitinib in TKI-resistant NSCLC. By lowering EGFR phosphorylation and raising EGFR degradation, curcumin causes apoptosis in TKI-resistant NSCLC cells, hence preventing the formation of cancer [151]. Most significantly, curcumin and EGFR-TKI therapy together significantly suppresses the development of NSCLC by lowering EGFR, c-MET, and cyclin D1 expression. Through the regulation of mitogen-activated protein kinase activity, the combination therapy improves intestinal epithelial cells’ survival rate and reduces intestinal mucosal damage [152].

In addition, as compared to curcumin or gefitinib therapy alone, the combination of the two induces significant autophagy activation, autophagic cell death, and autophagy-mediated apoptosis. Autophagic cell death brought on by therapy is lessened by pharmacological autophagy inhibitors such as bafilomycin A1 or 3-MA, beclin-1 or autophagy-related 7 knockdown, or both [152]. Frontline therapy with EGFR-TKIs includes afatinib and erlotinib. When erlotinib and curcumin are administered together, they cause apoptosis and increase IκB expression, which limits nuclear factor-κB (NF-κB) to the cytoplasm and inhibits its capacity to bind DNA. This drastically reduces the viability of NSCLC cells [153].

By reducing EGFR, surviving expression, and blocking NF-κB activity in erlotinib-resistant NSCLC cells, the combo therapy also markedly promotes apoptosis [154]. There is already a patent for the use of curcumin and afatinib together to treat NSCLC that is resistant to gefitinib and erlotinib (CN105476996A). For NSCLC, curcumin also reverses chemotherapy resistance. HIF-1α has been linked, in a recent study, to the development of chemotherapeutic resistance in cancer; consequently, cisplatin resistance may be reversed by targeting HIF-1α using RNA-interference or small interfering RNA. Combining curcumin with cisplatin has been shown to significantly reduce the growth of cisplatin-resistant NSCLC cells and induce apoptosis by activating caspase-3 and encouraging HIF-1α degradation, respectively [155].

In NSCLC, curcumin also overcomes cisplatin resistance by promoting cisplatin-induced apoptosis through the production of intracellular reactive oxygen species (ROS) and the proteosomal breakdown of Bcl-2 [156]. Studies on androgen-dependent and androgen-independent prostate cancer cells have shown that curcumin can suppress EGFR activity [157]. Curcumin and beta-phenyl ethyl isothiocyanate (PEITC) treatment caused the prostate cancer line PC-3 to undergo apoptosis, resulting in caspase-3 disruption and the suppression of EGFR- and EGF-induced Akt and PI3K activation [158]. Some clinical phase trials linked with curcumin in lung cancer are provided in the Table 3.

#### 13.2.4. Fisetin

With a diphenylpropane structure, fisetin (3,3′,4′,7-tetrahydroxyflavone) is a naturally occurring bioactive flavonol. Apples, strawberries, cucumbers, persimmons, and a variety of acacia plants and shrubs are common sources of fisetin. Its anti-inflammatory, anti-microbial, anticancer, and neuroprotective qualities have been demonstrated previously [162]. By modifying a number of cell signaling pathways, such as inflammation, apoptosis, angiogenesis, growth factor, transcription factor, and others, fisetin has been shown to have anticancer properties [163]. In vitro tumor angiogenesis and VEGFR expression in Y79 cells were both dose-dependently reduced by fisetin. Fisetin may, therefore, be a viable treatment option for retinoblastoma angiogenesis inhibition since it was discovered to do so by blocking the VEGF/VEGFR signaling pathway [102]. Fisetin enhanced the cytotoxic effects of erlotinib, a TKI, and reduced the capacity of H1299 cells to establish colonies on soft agar [164]. Fsetin’s antiproliferative effect, which entails causing a modest G2/M arrest and stopping the cell cycle in the G1 phase, is ascribed to its inhibition of VEGF production [165]. In non-small cell lung cancer, a novel 4′-brominated derivative of fisetin inhibits the EGFR/ERK1/2/STAT3 pathways and causes cell cycle arrest and apoptosis without having any negative effects on mice. Analogs of fisetin also inhibited the EGFR/ERK1/2/STAT3 pathways. A greater ratio of Bax to Bcl-2 expression was seen in conjunction with apoptosis triggered by fisetin analogue [166].

#### 13.2.5. Formononetin

One of the main biomolecules extracted from red clover and the Chinese plant *Astragalus membranaceus* is formononetin, a methoxylated isoflavone (7-hydroxy-3-(4-methoxypheny)-4H-1-benzopyran-4-one), which is a member of the isoflavonoid group of phytoestrogens. Formononetin exerts anticancer effects by modulating several cellular functions and molecular signaling pathways in various malignancies [167]. In NSCLC cell lines (HCC827 (EGFR Del E746-A750), H1975 (EGFR L858R/T790M), H3255 (EGFR L858R), A549 (EGFR WT), and H1299 (EGFR WT), formononetin was shown to promote the efficacy of EGFR-TKI by modulating the EGFR-AKT-Mcl-1 axis in a ubiquitination-dependent manner [101]. According to in vitro research using various human cancer cell lines, formononetin inhibits the development of carcinogenesis and metastasis by several mechanisms, such as controlling transcription factors, altering epigenetic targets, controlling estrogen receptors, controlling the cell cycle, triggering apoptosis, and controlling growth and developmental signaling pathways [168].

Inspired by the binding mechanism of lapatinib to EGFR, several novel formononetin compounds were designed and synthesized. Compound 4v showed the strongest anti-EGFR and anti-proliferation effect against the MDA-MB-231 cell line, which was comparable to that of lapatinib, according to an in vitro EGFR and cell growth inhibition experiment. The findings of additional biological experiments showed that 4v could effectively target EGFR and, subsequently, inhibit downstream signaling pathways, including EGFR/PI3K/Akt/Bad, EGFR/ERK, and EGFR/PI3K/Akt/β-catenin, respectively, in order to induce apoptosis, limit proliferation, and migrate in MDA-MB-231 cells [169].

#### 13.2.6. Luteolin

Common flavones like luteolin (3,4,5,7-tetrahydroxyflavone) are present in foods including celery, carrots, peppers, thyme, oregano, and more. The benefits of luteolin are diverse and include anti-inflammatory, anti-oxidant, neuroprotective, cardio-protective, and anticancer properties [170]. It has been discovered that luteolin inhibits EGFR autophosphorylation and causes EGFR degradation through the lysosomal pathway in NSCLC (NCI-H1975) and epidermoid carcinoma (A431) cells, respectively [100,171]. Numerous RTK-related components, such as VEGF, PI3K, Akt, MAPK, MMP-2, MMP-9, focal adhesion kinase (Fak), B-cell lymphoma/leukemia 2 (Bcl-2), B-cell lymphoma extra-large (Bcl-xL), nuclear factor-kappa b (NF-κB), cyclooxygenase-2 (COX-2), signal transducer and activator of transcription 3 (STAT-3), tumor necrosis factor-alpha (TNF-α), and hypoxia-inducible factor-1 alpha (HIF-1α), were all inhibited by luteolin [172,173]. By inhibiting the ROS system and VEGF-induced gastric cancer cell angiogenesis by regulating the VEGFR2 signaling pathway, luteolin lowers VEGF expression [174].

The acquired resistance of first- and second-generation EGFR-TKIs caused by the EGFR T790M mutation in NSCLC has been overcome by osimertinib, a third-generation EGFR-TKI. Osimertinib and luteolin together induced apoptosis and prevented H1975/OR cells from proliferating, migrating, and invading. By inhibiting the HGF-MET-Akt pathway, luteolin and osimertinib can work in concert to overcome MET amplification and overactivation-induced acquired resistance to osimertinib. This suggests that luteolin and osimertinib may be used clinically in patients with acquired resistance in non-small cell lung cancer [175].

A number of cellular processes that contribute to the growth and advancement of cancer cells can be inhibited by luteolin through the reduction in the activity of certain receptor tyrosine kinases (RTKs), including IGFR, EGFR, and ERs, as well as their downstream effector molecules [176]. The combined effects of luteolin and gefinitib, a selective tyrosine kinase inhibitor that suppresses both EGFR and the kinase activity of cyclin G-associated kinase (GAK), on PC-3 prostate cancer cells were investigated in 2014 [177].

#### 13.2.7. Quercetin

Quercetin is also known as 3,3,4,5,7-pentahydroxy-2-phenylchromen-4-one, reflecting the presence of five hydroxyl groups at positions 3-,5-,7-, and 4 of its molecular structure. Quercetin inhibits cancer development and progression through the modulation of various cell signaling pathways. Quercetin exhibits several pharmacological effects, including antibacterial, anti-inflammatory, anticancer, and antioxidant properties. Extensive research over recent decades has shown that quercetin possesses anti-ulcer, anti-allergy, antitumor, antiviral, and antidiabetic properties, along with anti-hypertension, anti-infection, gastro-protection, and immuno-modulation effects [178]. Quercetin has been documented to show a notable protective effect against metronidazole-induced neuronal damage [179].

Numerous studies have highlighted quercetin’s anti-allergic, anti-inflammatory, antiviral, and anticancer effects [180,181]. Quercetin has been shown to dramatically suppress the expression of growth factor signaling molecules, including EGFR, pAKT, pGSK-3β, β-Catenin, NFκB, and cyclin D, in lung cancer cells (A549) [182]. Additionally, after quercetin therapy, the study observed a reduction in matrix metalloproteinase 2 (MMP-2) and MMP-9 expression, which was probably caused by the EGFR/Akt/β catenin signaling pathway, and had an anti-metastatic impact. Molecular docking studies have extensively investigated quercetin’s ability to bind to receptor TKs (RTKs). Quercetin has been demonstrated to bind via hydrogen bonds, hydrophobic contacts, and π-π interactions to the ATP binding pocket or active site of RTKs such as EGFR, VEGFR2, FGFR1, IGF1R, and c-MET [183,184].

Quercetin targets several kinases involved in the growth and progression of cancerous cells. Quercetin inhibits the PI3K-Akt/PKB pathway by binding to PI3Kγ (IC50 = 3.8 µM) without targeting Akt/PKB [185,186]. As a strong inhibitor of TKs, including Syk, Src, Fyn, and Lyn [187], quercetin is the most investigated flavonoid. It has additionally been demonstrated to have anti-metastasis properties on stomach cancer cell lines [188]. It was demonstrated, in a phase I clinical trial, that intravenous quercetin had an anticancer effect in cisplatin-resistant ovarian cancer, evidenced by its antiproliferative effect and the inhibition of lymphocyte TK activity [103].

### 13.3. Phenolic Molecules

#### 13.3.1. Caffeic Acid

All plant species produce caffeic acid, a phenolic molecule that has anti-inflammatory, anti-carcinogenic, and antioxidant properties. It can be found in drinks like coffee, wine, and tea as well as in common medications like propolis [189]. By increasing ROS levels and compromising mitochondrial function, caffeic acid can cause cancer cells to undergo apoptosis. Caffeic acid and its derivatives in cancer therapy have an impact on molecular pathways that play a part in the progression of cancer, including PI3K/Akt and AMPK. By blocking the epithelial-to-mesenchymal transition process, caffeic acid suppresses metastasis and lessens the aggressive aggressiveness of malignancies. Notably, caffeic acid and caffeic acid phenethyl ester (CAPE) can increase cancer cells’ sensitivity to chemotherapy-induced cell death and their response to chemotherapy. Caffeic acid and CAPE have been used in combination with other antitumor agents such gallic acid and p-coumaric acid to increase their ability to suppress malignancy. Because of its low bioavailability, nanocarriers have been created to increase its capacity to suppress cancer [190].

In the therapy of cancer, it has been discovered that caffeic acid targets receptor tyrosine kinases (RTK). One type of RTK is the cell-surface receptor for epidermal growth factor, known as the EGFR. In breast cancer cells, caffeine inhibits the phosphorylation of EGFR [191]. By modifying important genes and proteins involved in cancer resistance and treatment, CAPE is a therapeutically useful chemical that helps AZD9291 treat EGFR-TKI-resistant cells [104]. CAPE and docetaxel treatment together significantly reduced the expression of SKP2, c-MYC, and phospho-EGFR (Tyr 992) proteins in NSCLC compared to either CAPE or docetaxel treatment alone [192].

#### 13.3.2. Epigallocatechin-3-Gallate (EGCG)

Epigallocatechin-3-gallate (EGCG), the primary active component of green tea, has been shown to have preventive and therapeutic effects against various diseases. The health benefits are mainly attributed to its potent anti-inflammatory and antioxidant qualities. EGCG’s anticancer effects have been observed in multiple types of cancer and are still being investigated. It has been demonstrated that EGCG has a chemopreventive impact by blocking the initiation, promotion, and development of the carcinogenesis process [193]. EGCG, the most abundant catechins of green tea, has been widely researched in several studies for its anti-carcinogenic properties [194]. The bioavailability of chemopreventive drugs, such as EGCG, has been increased by the application of recently developed nanotechnology [195,196].

In order to help cure human lung cancer cells, an EGCG nanoemulsion (nano-EGCG) was created. The anticancer effects of this emulsion were systematically investigated. Previous studies suggest that EGCG exhibits a number of biological effects, including anti-inflammatory, antitumor, antidiabetic, and anti-obesity effects [194]. EGCG has varying effects on the few NSCLC cell lines that have been studied, despite its ability to stop the growth of small cell lung cancer cells [197]. In addition to causing cell death, EGCG may interfere with angiogenesis, migration, invasion, carcinogenic activity, and tumor formation. In vitro and in vivo studies indicate that EGCG effectively induces apoptosis and inhibits the growth in a number of cancers, including leukemia and brain, breast, kidney, and colon tumors [105]. These antitumor effects are associated with the modulation of several signaling molecules, including reactive oxygen species, NF-κB, Akt, vascular endothelial growth factor, peroxisome proliferator-activated receptor, Bcl-2, and mitogen-activated protein kinases, as well as epigenetic modification [198]. While EGCG inhibits the growth of small cell lung cancer cells, it exhibits inconsistent effects on the small number of NSCLC cell lines tested [199,200]. EGCG causes EGFR internalization and ubiquitin degradation, leading to the disruption of EGFR signaling, whereas erlotinib prevents EGFR phosphorylation and maintains EGFR at the plasma membrane [201].

#### 13.3.3. Ellagic Acid

Ellagic acid (EA) is a thermostable polyphenolic molecule naturally occurring in a wide range of fruits and nuts, such as black currants, raspberries, strawberries, walnuts, and grapes. EA exists either in a bound form, such as ellagitannins, or in its free form like glycosides. It exhibits many advantageous pharmacological properties such as antidiabetic, antimutagenic, antibacterial, antiviral, anticancer, and chemoprotective effects [202]. EA has been found to be a naturally occurring dual inhibitor of VEGF and PDGF receptors, indicating that it has significant antiangiogenic qualities that could aid in the prevention and management of cancer [111]. Erlotinib and EA work in concert to combat the EGFR H773_V774 insH mutation [203]. The dual inhibitor role of EA suggests that EA can be used as a leading promising compound for the development of novel plant-derived TKIs in cancer therapy.

#### 13.3.4. Gossypol

Gossypol is a polyphenol compound found in the seeds of cotton plants (*Gossypium* sp.) and is known for its defensive properties. It has been investigated for the treatment of gynecological diseases as well as a potential oral male contraceptive. Extensive research has documented its antitumor, antioxidant, antiviral, antibacterial, and immunomodulatory properties [204]. Gossypol exerts its anticancer action by inhibiting anti-apoptotic Bcl-2 family proteins, thereby promoting apoptosis [112]. Gossypol has demonstrated its anticancer activity across multiple human breast cancer cells (MCF-7, MDA-MB-231, MDA-MB-468, ZR-75-1, and T47D), pancreatic cancer cells (BxPC-3 and MIA PaCa-2), human colon cancer cells (COLO 225), human cervical cancer cells (HeLa and SiHa cell lines), non-small cell lung cancer (NSCLC) cell lines (H1975), human lung cancer cell lines (H1299 and H358), and prostate cancer cells [205,206].

Although a number of EGFR inhibitors have been used to treat NSCLC, their effectiveness has decreased due to the introduction of EGFRL858R/T790M resistance mutations. The EGFR T790 M mutation is the primary mechanism of EGFR-TKI resistance and complicates treatment in NSCLC patients with activating EGFR mutations. Additionally, resistance to EGFR-TKIs is conferred by YAP/TAZ. By targeting both YAP/TAZ and EGFRL858R/T790M, gossypol emerges as a promising therapeutic candidate for overcoming EGFR-TKI resistance [206]. Gossypol inhibited cell proliferation and induced the apoptosis of NSCLC cells by targeting EGFRL858R/T790M mutation [207].

#### 13.3.5. Honokiol

Honokiol is a biphenolic compound isolated from *Magnolia* species. It exhibits significant anticancer activity in NSCLC. Honokiol has been documented to downregulate the P13K/AkT/mTOR signaling pathways by decreasing AkT phosphorylation and upregulating pTEN expression. Thereby, it impairs cancer cell survival and proliferation [113]. Moreover, honokiol inhibits STAT3 phosphorylation irrespective of EGFR mutation status, suggesting broad-spectrum efficacy. The critical role of STAT3 in mediating the effects of honokiol is further underscored by findings that STAT3 knockdown abolishes honokiol-induced anti-proliferative and anti-metastatic activities [114]. Through simultaneously targeting STAT3 and P13K/AkT/mTOR pathways, honokiol represents a multitargeted agent for combating NSCLC progression and metastasis.

#### 13.3.6. Magnolol

Magnolol is a neolignan and is derived from *Magnolia officinalis*. Magnolol has shown potent anticancer activities against various types of cancers including NSCLC. It exerts its effects by targeting multiple signaling pathways relayed by TKs, particularly the EGFR/P13K/Akt/mTOR and MAPK/ERK axes, which are frequently dysregulated in NSCLC. By blocking EGFR phosphorylation, restoring PTEN, and activating caspases, it effectively suppresses tumor growth and overcomes resistance, highlighting its potential as a supportive therapy in TKI refractory lung cancer [208].

### 13.4. Stilbene

#### Resveratrol

Resveratrol is a naturally occurring stilbene that offers several beneficial medicinal properties. Certain plants, including a variety of food sources such as peanuts, plums, apples, raspberries, blueberries, grapes, and goods made from them, are important sources of resveratrol. Many plants produce resveratrol in response to injury [209], and it has been discovered that over 70 plant species contain this natural polyphenol [210]. Resveratrol has been demonstrated to possess a number of therapeutic properties, such as powerful antioxidant activity and anti-inflammatory and antiproliferative properties [211,212,213].

Hydroxyl derivatives like oxyresveratrol exhibit similar effects [214,215]. In order to overcome gefitinib resistance, resveratrol increases intracellular gefitinib concentration, inhibits the removal of gefitinib from cells, and reduces CYP1A1 and ABCG2 expression. When gefitinib accumulates in gefitinib-resistant NSCLC cells, it causes apoptosis, autophagy, and senescence [216]. The anti-NSCLC actions of gefitinib are consequently enhanced when co-treated with resveratrol. Although it enhances the efficacy of TKIs, the direct inhibition of TKs by resveratrol has not been clearly documented.

### 13.5. Saponins

#### 13.5.1. Ginsenosides

In Asian nations, *Panax ginseng*, commonly referred to as Korean ginseng, is a traditional medicine. Triterpenoid saponins, which are ginsenosides, are its main active ingredients. Among these, Re, a well-known ginsenoside, has demonstrated a number of biological actions, such as anti-inflammatory and anticancer qualities. *Panax ginseng*, a member of the Araliaceous plant groups, has long been utilized in both medicine and cooking. In East Asian countries like China and Korea, *Panax ginseng* has been used for preventing as well as treating cancer and other diseases [217,218].

In ginseng roots, fruits, stems, and leaves, ginsenosides are the primary saponins. To date, approximately 60 ginsenosides have been identified and isolated. Ginsenosides share a common core structure, but different sub-types vary in the arrangements of the four steroidal rings, including the A-Panaxadiol group (e.g., Rb1, Rb2, Rb3, Rc, Rd, Rg3, and Rh2), the B-Panaxatriol group (e.g., Re, Rg1, Rg2, and Rh1), and the C-Oleanolic acid group (e.g., Ro) [219].

Additionally, in NSCLC, ginsenosides enhance gefitinib’s anticancer action. In NSCLC cells, ginsenoside Rg3 has been shown to enhance the cytotoxic effects of gefitinib in a way that is dependent on both time and dosage. Gefitinib and ginsenoside Rg3 together dramatically boost apoptosis in non-small cell lung cancer cells by upregulating the pro-apoptotic protein Bax and caspase-3 activity while downregulating the anti-apoptotic protein Bcl-2 [220]. As master regulators that control several genes involved in EMT, such as E-cadherin [221], the combination therapy also prevents NSCLC cell migration by reducing the protein expression of Snail and Slug [220]. According to a study, ginsenosides inhibited the growth of NSCLC cells by activating the tumor suppressor p53-binding protein-1 and upregulating vaccinia-related kinase 1 [119]. Osimertinib is a third-generation EGFR-TKI, which has been associated with increased stemness and a higher potential for metastasis in osimertinib-resistant NSCLC cells compared with their parental cells [222].

Remarkably, ginsenoside Rg3 reduces the stemness of NSCLC cells and their resistance to osimertinib. This is evidenced by decreased stemness marker expression and reduced spheroid formation, which is reliant on the Hippo signaling pathway [118]. Ginsenosides also help patients who are resistant to chemotherapy medications like cisplatin (DDP), which is used to treat non-small cell lung cancer. Clinical studies demonstrate that ginsenoside Rg3 improves chemotherapy’s therapeutic effects in NSCLC patients. The patients receiving Rg3 therapy had longer overall survival and improved quality of life [223].

The leucocyte count is restored, the CD4/CD8 T-cell ratio is raised, and vascular endothelial growth factor (VEGF) expression is decreased in these individuals by Rg3. Many clinical studies show that combining Rg3 with chemotherapy improves the efficacy and overall survival time of patients with NSCLC. Ginsenoside may also help NSCLC patients overcome their cisplatin resistance, according to experimental studies. Ginsenoside suppresses the NRF2 pathway and makes A549/DDP cells much more sensitive to cisplatin in cisplatin-resistant NSCLC cells. NRF2 knockdown reduces ginsenosides’ synergistic effects in cells resistant to cisplatin [224]. According to prior research, Rg3 also reduced PD-L1 expression by inhibiting NF-κB p65 and Akt, which restored T cell cytotoxicity against cancer cells and reduced A549/DDP cells’ resistance to cisplatin therapy [225]. Although ginsenosides show numerous therapeutic responses, they do not directly interact or inhibit TKs, and their anticancer effects are mediated by the modulation of key cell signaling pathways.

#### 13.5.2. Astragaloside IV

The decoctions made from the roots of *Astragali Radix* (Huangqi), are widely used in traditional Chinese medicine to treat cancer, inflammation, and bacterial and viral infections. Astragaloside IV (AS-IV), a cycloartane-type triterpene glycoside, is the primary active compound in *Astragalus membranaceus* and is responsible for its pharmacological effects. AS-IV serves as a standard marker for quality control in the Chinese pharmacopoeia. The primary active ingredient of *Astragali Radix*, AS-IV, can prevent cardiovascular disease [226], protect the liver [227], treat antidiabetic nephropathy [228], and have anticancer properties [120]. AS-IV enhances the sensibility of lung adenocarcinoma cells to bevacizumab by inhibiting autophagy [229]. Due to its demonstrated efficacy in a variety of cancer models of colorectal [230], liver [231], and lung [229] carcinomas, AS-IV is being explored as a promising anticancer agent. AS-IV can also be utilized to boost the sensitivity of chemotherapeutic medications and in conjunction with other anticancer drugs [232].

AS-IV has shown a good safety profile and low toxicities in experimental studies. By controlling sirtuin 6 (SIRT6), AS-IV sensitizes NSCLC to gefitinib. Inhibiting SIRT6 eliminates the sensitization effect of AS-IV in NSCLC cells, such as HCI-H1299, HCC827, and A549 cells [233]. SIRT6 is an NAD-dependent deacetylase that facilitates NSCLC. The inhibition of transforming growth factor-β1-induced EMT in NSCLC cells by SIRT6 depletion highlights how astragaloside IV enhances gefitinib’s anti-NSCLC activities. Recent studies support the potential of AS-IV as a novel anticancer therapeutic medication, as demonstrated by recently published studies showing strong antitumor efficacy [234]. However, no current report supports the direct inhibition of TKs by AS-IV, and it is grouped to be a bioactive modulator with a significant ability for the complement of TKI therapy.

#### 13.5.3. Polyphyllin

The primary bioactive ingredient in *Paris polyphylla* is polyphyllin I (PPI). Recent studies on PPI across a variety of cancer types have demonstrated that PPI exerts diverse antitumor effects, including inducing cell cycle arrest, promoting cell apoptosis, triggering autophagy, inhibiting angiogenesis, enhancing tumor sensitivity to chemotherapy, and modulating immune and inflammatory responses [235]. PP I demonstrates strong antitumor action in both in vitro and in vivo models and increases the sensitivity of drug-resistant cells to gefitinib. PP I induces NSCLC cell death through multiple mechanisms, including the activation of the SAPK/JNK pathway, the suppression of p65 and DNMT1 expression [122], the downregulation of MALAT1, and the inhibition of the suppression of STAT3 phosphorylation [123]. PP I also reverses epithelial–mesenchymal transition (EMT) in HCC827 cells by inhibiting the prevention of IL-6 signaling, thereby overcoming resistance to erlotinib [236]. By blocking the PI3K/AKT/mTOR pathway, PP II can stop the proliferation and induce the apoptosis of gefitinib-resistant PC-9/ZD cells [237].

### 13.6. Triterpenes

#### 13.6.1. Cucurbitacin

Cucurbitacin and its derivatives are categorized into 12 groups, including dihydrocucurbitacin B, A, B, C, D, E, I, H, Q, and R, with 40 related compounds currently recognized [238]. Among them, cucurbitacin B, a steroid with a distinct bitter taste and cytotoxic qualities, has been most widely investigated. Cucurbitacin B (C32H46O8, molecular weight 558.712 g/mol, 19-(10➝9β)-abeo-10-lanost-5-ene triterpene) is present in plants of the cucurbitaceae family as well as other plant families like Brassicaceae. Its bitterness may serve as a natural defense against parasites and predators [239].

It has shown efficacy against a number of medical conditions, including immune-related and angiogenic disorders, cell adhesion and leukemic disorders, insect infestation, carbon chloride-induced hepatotoxicity and profound cholestasis, generalized inflammation and algesia, and CD18-mediated disorders [240,241]. A chemical analog of cucurbitacin B was isolated from fruiting bodies of a mushroom “*Leucopaxillus gentianeus*” in another study. This analog differed from cucurbitacin B in that it lacked an oxygen substituent at carbon-16. It was then screened for antitumor toxicity based on variations in its chemical structure [242].

Through hydrophobic and electrostatic forces, cucurbitacin B binds strongly with human serum albumin and increases ibuprofen–albumin binding [243]. Cucurbitacin B rapidly altered breast cancer cell morphology by disrupting F-actin and microtubules within 15–20 min after exposure in a dose-dependent manner. These results were confirmed in naked mice, where cucurbitacin B injections administered intraperitoneally at a dose of 1 mg/kg every three weeks produced a suppression of tumor size growth of around 50% [244].

By triggering G2/M cell-cycle arrest and mitochondrial death, cucurbitacin B effectively inhibits the development of non-small cell lung cancer. The thiol antioxidant N-acetyl cysteine reduces the anti-NSCLC effects of cucurbitacin B, whereas butithione-sulfoxime, an inhibitor of glutathione formation, intensifies the anti-NSCLC effects [245]. According to another study, cucurbitacin B treatment significantly decreases the ratio of thiol and glutathione to oxidized glutathione in NSCLC cells, demonstrating that the anti-NSCLC properties of cucurbitacin B are facilitated by an impairment of the cellular redox balance [246].

In NSCLC cells, cucurbitacin B and D also conquer gefitinib resistance. By dephosphorylating oncogenic kinases and transcription factors, CIP2A hinders protein phosphatase-2A (PP2A) [247], leading to a reduction in tumor development [248]. Additionally, in NSCLC cells, the lysosomal degradation of EGFR is induced by the cucurbitacin-B-mediated suppression of the CIP2A/PP2A/Akt axis [249]. In a prior work, a solid-phase EGF-EGFR interaction test showed that cucurbitacin D overcomes gefitinib resistance by preventing EGF from binding to EGFR. Cucurbitacin D was found to directly disrupt the interaction between EGFR and EGF. As a result, cucurbitacin D reduces EGFR phosphorylation in gefitinib-resistant NSCLC cells [250].

#### 13.6.2. Betulinic Acid and Betulin

Betulinic acid (BA), a pentacyclic triterpene molecule, can be extracted from Birch tree bark and it can also be chemically synthesized or modified through biotransformation [251]. BA exerts anticancer effects by binding to specific target molecules and modulating key cellular signaling pathways [252,253,254]. The primary anticancer mechanism of BA involves the induction of mitochondrial-mediated apoptosis [255,256,257] with no notable side effects [258,259]. BA has been reported to exert several biological effects, including the modulation of transcription factors [260,261], suppression of NF-κB and STAT3 signaling pathways [262,263], and regulation of autophagy [256].

Due to its limited water solubility, BA remains under investigation and is not yet widely used in clinical oncology. Numerous biological activities, including antitumor, anti-inflammatory, anti-HIV, antimalarial, antibacterial, and antioxidant properties, are exhibited by BA and its derivatives (produced by modification at the C-3, C-20, and C-28 sites) [264,265,266,267]. It is possible that BA works better when combined with other anticancer medications. For example, BA suppresses the growth of the pancreatic cancer cells (PANC-1 and Mia PacA-2) by activating the AMPK pathway. Patients with lung adenocarcinoma who have an EGFR mutation are treated with EGFR-TKIs, such as erlotinib and gefitinib, as their first line of treatment. However, drug resistance typically develops within 8–11 months of treatment. Co-treatment with BA and either erlotinib or gefitinib significantly reduced the viability of drug-resistant H1975 cancer cells in a dose-dependent manner in comparison with erlotinib or gefitinib alone and increased the apoptotic cell populations [268,269]. BA was more successful in preventing PANC-1 cell growth and triggering apoptosis when coupled with gemcitabine [270]. BA exhibits cytotoxic effects that contribute to tumor suppression via kinase-dependent mechanisms, although it does not directly inhibit TK activity.

Betulin is a Triterpenoid and is found in many plants and herbs. In the human chronic myelogenous leukemia cell line, betulin modulates the mitogen-activated protein (MAP) kinase pathway with activity comparable to that of the well-known ABL1 kinase inhibitor imatinib mesylate [127].

#### 13.6.3. Leelamine

Because of their distinct mode of action and variety of molecular structures, natural molecules are propelling the creation of novel TKIs. New advancements in the study of anticancer drug discovery are encouraged by the identification of novel TK inhibitory molecules from natural products [271]. Pine bark plants naturally produce a lysosomotropic chemical called leelamine (LEE) or dehydroabietylamine, which is a lipophilic diterpene amine. Numerous earlier investigations have revealed LEE’s possible anticancer properties. LEE was reported to modulate the STAT5 pathway in myelogenous leukemia cells, causing both autophagy and death. It has also been documented that LEE can activate PTPε in multiple myeloma cells and block the STAT3/JAK signaling pathway [128]. LEE prevents intracellular cholesterol transfer. Its lysosomotropic properties, accumulation in the lysosome, and blockage of cholesterol transport result in a lack of availability for essential activities needed for cancer cell activity, which, in turn, causes cell death [272]. Interestingly, the pyruvate dehydrogenase kinase inhibitors (PDK inhibitors) used in all of the studies looking into combination therapy including EGFR-TKIs and PDK inhibitors to overcome resistance in NSCLC were synthetic. Huzhangoside A, leelamine, and otobaphenol are examples of natural-product-based PDK inhibitors that have been shown in a prior study to promote pyruvate dehydrogenase (PDH)-activity-dependent cancer cell death [273].

## 14. Translational Insights into Natural-Product-Enhanced Lung Cancer Combinational Therapy

Chemotherapeutic drugs may have one or several adverse effects; therefore, there is a growing interest in the combination of chemotherapeutic drugs with fewer or minimal toxic natural molecules for the management of tumors. In order to improve tumor treatment, the therapeutic approach of combining natural compounds with chemotherapeutic drugs can lessen the development of drug resistance in tumor cells, improve the tumor-killing effect of chemotherapeutic drugs, and lessen the severe side effects of chemotherapeutic drugs on patients [274].

The viability of using EGFR-TKI in conjunction with other NSCLC treatments, such as radiation, cytotoxic chemotherapies, targeted therapies, and new immunotherapies, has been the subject of numerous clinical investigations. However, the development and application of these solutions are hampered by the substantial gap that still persists when extrapolating preclinical outcomes to clinical circumstances [275]. The ideas of synergy and combination therapy are closely related, and it has been noticed that the quality of life for patients with advanced diseases like cancer, hypertension, and asthma has been significantly improved by evolving combination therapy [276]. The combination of phytocompounds or plant extracts with complementary therapy has been widely valued, and it has been used for the treatment of cancer. It has been suggested that the specific properties of natural substances and their capacity for enhancing the effects of chemotherapeutic drugs may improve lung cancer patient outcomes and may open the door for more individualized and efficient future treatment options [277].

It has been documented that natural products reduce chemotherapy-induced tumor resistance. Hence, when used in conjunction with chemotherapy, natural products may work in concert to increase antitumor effects and may encourage tumor cell death [278]. Moreover, there are several mechanisms of action of natural products, like sensitizing cancer cells to increase the tumoricidal effect, reversing chemoresistance by inhibiting multiple targets involved in the development of drug resistance, and reducing chemotherapy-induced toxicity in non-tumoral cells by encouraging repair mechanisms, which may contribute to their synergistic effects [279].

For lung cancer, the use of natural substances in conjunction with traditional chemotherapy medications has also shown encouraging outcomes [280]. Emodin has shown synergistic benefits with enhanced anticancer activity against lung adenocarcinoma and non-small cell lung cancer when combined with the chemotherapies like sorafenib, afatinib, cisplatin, paclitaxel, gemcitabine, and endoxifen [280,281]. When paired with EGFR-TKIs, a variety of natural compounds that have strong inhibitory effects on drug-resistant cells can greatly boost the sensitivity of these cells to EGFR-TKIs [86].

Additionally, *Scutellaria baicalensis* and cisplatin have shown synergistic effects, preventing tumor growth in vitro (Lewis lung carcinoma cells, or LLC) and in vivo (C57BL/6J tumor-inoculated mouse model) [282]. Using H460, H1975, and H292 cell lines, the extract from *Marsdenia tenacissima* and gefitinib worked in concert to cause apoptosis in resistant cells. The combo treatment decreased EGFR downstream signaling pathways, according to the results of flow cytometry [283]. Through the SHH signaling system, combination therapy with sulforaphane and gefitinib dose-dependently decreases the growth of gefitinib-resistant lung cancer cells and inhibits the expression of SHH, SMO, and GLI1 [284].

Solamargine has been reported to reduce cell proliferation and increase apoptosis in the cisplatin-resistant lung cancer cell lines NCI-H1299 and NCI-H460. More significantly, solamargine and cisplatin worked in concert, each increasing the effectiveness of the other [285]. Hederagenin is a pentacyclic triterpenoid that has been shown to increase the conversion of LC3-I to LC3-II in lung cancer cells, thereby inhibiting autophagy. It has also been suggested that hederagenin and paclitaxel and cisplatin, respectively, may work in concert to increase their anticancer effects [286]. By dose-dependently inhibiting TMEM16A, narirutin, a flavonoid extracted from citrus unshiu, enhanced the anticancer activity of cisplatin when used in conjunction with cisplatin for lung cancer [287].

In fact, an effectiveness investigation in NSCLC cells shown that the medication combination of cisplatin and curcumin had better benefits [288]. By controlling the Cu-Sp1-CTR1 regulatory loop, curcumin improved the therapeutic efficacy of cisplatin in lung cancer cell lines A549, H460, and H1299, according to an in vitro study [289]. It has been reported that extracts from *Terminalia bellerica* and *Phyllanthus emblica* function in combination with doxorubicin and cisplatin to prevent the growth of human lung cancer and hepatocellular carcinoma cells [290]. The combination of 5-Demethylnobiletin and paclitaxel substantially inhibited the proliferation of lung cancer cell lines [291].

In NSCLC Aloe-emodin may increase PC9-GR cells’ sensitivity to gefitinib and reverse epithelial–mesenchymal transition (EMT) via inhibiting the PI3K/Akt/TWIS1 signal pathway [292]. When compared to TKI alone, a combination of TKI and Chinese herbal medicines increased PFS in patients with advanced NSCLC that had an EGFR mutation [293]. Yiqi Chutan Tang (YQCT) boosts gefitinib-induced apoptosis, increases autophagy, causes mild cell cycle arrest, and decreases gefitinib-induced drug resistance. These findings suggest that, when combined with gefitinib, YQCT can boost anticancer effects and decrease drug resistance at the molecular level [294].

Anticancer TKIs (nilotinib, bosutinib, dasatinib, and ponatinib) and a methoxyflavanone derivative synergistically inhibit the growth of A549 cells. When combined with nilotinib and PDMF, they caused G2/M cell cycle arrest and de novo G1 arrest. In athymic nude mice, the co-administration of PDMF and nilotinib dramatically reduced the in vivo tumorigenicity of A549 cells [295]. When paired with osimertinib, berberine selectively and synergistically reduced the lifespan of many MET-amplified osimertinib-resistant EGFR-mutant NSCLC cell lines. This was likely due to increased apoptotic induction brought on by Bim elevation and Mcl-1 reduction. Crucially, this combination was well tolerated and successfully increased the suppressive effect on the growth of MET-amplified osimertinib-resistant xenografts in nude mice [296]. The combinational therapies of TKIs or chemotherapeutic drugs and natural products in lung cancer treatment are provided in Table 4.

## 15. Fourth-Generation TKIs

The development of acquired resistance to EGFR-TKIs is inevitable, which restricts their long-term efficacy. EGFR secondary mutations and bypass signaling activations are the primary mechanisms of the resistance process. A lot of work has gone into creating new fourth-generation EGFR-TKIs, even though first-, second-, and third-generation EGFR-TKIs have achieved great strides in NSCLC-targeted therapy. Allosteric inhibitors, such as EAI045, JBJ-25-02, JBJ-09-063, BLU-945, BLU-701, BDTX-1535, BBT-176, JIN-A02, TRX-221, BPI-361175, and BAY 2927088, have been discovered through preclinical drug development. These inhibitors bind to the EGFR allosteric site and cause conformational changes that reduce its affinity with ATP [86]. In order to overcome on-target resistance, fourth-generation TKIs are becoming a more intriguing therapeutic alternative. Thiazole-amid inhibitors, whose activity is mediated by the allosteric inhibition of EGFR and has a high selectivity towards mutant-EGFR, have been discovered as a result of preclinical therapeutic research. To clarify their activity in the therapeutic context as well, early phase 1/2 clinical trials are now being conducted [297]. As a fourth-generation EGFR-TKI, BLU945 selectively targets co-mutations in T790M and C797S as well as other T790M resistance mutations brought on by osimertinib resistance [298]. Although TQB3804 is undergoing phase I (NCT04128085) and phase II (NCT04180150) trials, no pertinent data have been made public as of yet; therefore, its clinical potential will have to be ascertained later. The BBT-176 was unable to avoid the fate of halting research and development, as evidenced by the termination of its clinical study. In the preclinical stage, the same company’s BBT-207, which is now being developed (NCTNCT05920135), has demonstrated outstanding antitumor effects against double mutations, including C797S, and triple mutations [299].

## 16. Caspase-Mediated Apoptosis Induced by Phytochemicals in Lung Cancer Models

Apoptosis or cellular death is caused by the proteolytic cleavage of millions of proteins by effector caspases, and it is a crucial mechanism for tumor suppression [300]. Multicellular organisms depend on cell death for proper development, tissue homeostasis, and integrity. Various executioner and regulatory molecule groups carefully govern apoptosis. Cell shrinkage, dynamic membrane blebbing, the condensation of chromatin material, DNA fragmentation in the nucleus, and the loss of adherence to the external matrix are characteristic mechanisms of action of apoptotic cell death. Additional biochemical changes include phosphatidylserine externalization and the activation of cysteine aspartyl proteases, or caspases, which cause cell death [301]. Although certain drugs may induce apoptosis through caspase-independent routes, caspases are crucial mediators of apoptotic signals from upstream molecules, and their activation is thought to be a hallmark of apoptosis [302].

Numerous diseases, including cancer, autoimmune diseases, neurological diseases, and cardiovascular diseases, are linked to the dysregulation of apoptosis. In apoptotic signaling cascades, initiator caspases (caspase-8 and -9) and executioner caspases (caspase-3, -6, and -7) are essential. Tumor necrosis factor (TNF) and other ligands start the extrinsic route by generating a death-inducing signaling complex (DISC), which activates caspase-8 and subsequent caspases [303].

The incapacity to undergo apoptosis in response to apoptotic stimuli is a hallmark of human cancers. It has been discovered that the disruption of initiator caspase-8 and caspase-9 expression or function may play a role in the development or progression of cancer, and that their inactivation may increase resistance to existing therapeutic strategies [304]. In one study, it was documented that curcumin did not activate Bcl-2 or Bcl-xl-transfected cells, but it did activate caspase-8 and caspase-3 in HL-60 neo cells [305]. Natural products such as tea extracts are rich in flavonoids and phenolic compounds, which exhibit anti-proliferative effects. These extracts promote apoptosis in cancer cells by enhancing the activity of caspase-3, caspase-7, caspase-8, and caspase-9 [306].

A study explored the mechanism of *Hericium erinaceus* (Yamabushitake)-mushroom-induced apoptosis of U937 human monocytic leukemia cells. When compared to the vehicle-treated group, the HWE and MWE treatments significantly raised the activities of caspase-9 and caspase-3. There was also an increase in caspase-8 activity. However, compared to caspase-3 and -9, the induction ratio of caspase-8 activity was significantly lower [307]. In NB4 human leukemia cells, hispolon from *Phellinus linteus* causes G0/G1 cell cycle arrest and apoptosis. It also induces the expression of proteins linked to apoptosis, including the proapoptotic Bax protein, poly (ADP ribose) polymerase, and the cleavage form of caspase 3, caspase 8, and caspase 9 [308].

It was discovered that procaspase-6, procaspase-8, and procaspase-9 expression significantly decreased when cells were treated with Lupeol; procaspase-3 protein status remained unchanged. These findings imply that caspase-8 and caspase-9 mediate and caspase-6 carries out the death of prostate cancer cells triggered by luteol [302]. The biological activity of Thymus vulgaris (TVEO) and Thymus serpyllum (TSEO) essential oils differ. TVEO activated caspase-3 and caspase-8 to induce apoptosis in cervical cancer HeLa cells, whereas TSEO activated caspase-3 to induce apoptosis [309]. Through the activation of caspase cascades and the downregulation of the inhibitor of apoptosis proteins via JNK/p38 signaling, curcumin analog L48H37 causes apoptosis in human oral cancer cells [310]. The apoptosis of MCF7 was induced, followed by the activation of pro-apoptotic genes (p53, Bax, Parp, Caspase-3, -8, -9) and the inactivation of antiapoptotic genes, such as the Bcl2 gene, in a study involving the growth inhibition and apoptosis of cancer cells by the ethyl 4-[(4-methylbenzyl)oxy] benzoate complex in vitro and in vivo [311]. Both in vivo and in vitro, tanshinone IIA causes intrinsic apoptosis in osteosarcoma cells linked to mitochondrial malfunction. In xerograft mice and Tan IIA-treated 143B cells, apoptosis was linked to the activation of the caspase cascade through Bcl-2 family modification [312].

## 17. Role of RTKs and Its Inhibitors for the Development and Treatment of BRAF Mutant Lung Cancers

The BRAF gene encodes V-Raf murine sarcoma viral oncogene homolog B (BRAF) kinase, which is essential for cell signaling, growth, and survival. The onset and spread of cancer are caused by mutations in the BRAF gene. BRAF mutations are frequently found in women, never-smokers, and aggressive histological forms of non-small cell lung cancer (NSCLC). They also account for between 1% and 2% of adenocarcinomas. Patients with BRAF-mutated non-small cell lung cancer had limited response to traditional treatment. However, the way that NSCLC is treated has changed significantly with the introduction of immune checkpoint inhibitors (ICIs) and targeted therapy [313].

Based on the location of the mutation, BRAF mutations can be categorized into three types. Class II mutants, such as K601, L597, G464, and G469 alterations, are found in the activation region or P-loop and signal as RAS-independent dimers [314,315]. Class I mutants, such as V600E/K/D/R, occur in the valine residue at amino acid position 600 of exon 15 [314,316]. BRAF kinase function is compromised by class III mutants that arise in the P-loop, catalytic loop, or DFG motif; in fact, non-V600 mutations account for about half of BRAF mutations in non-small cell lung cancer (NSCLC) [317,318,319]. Furthermore, because class II and III BRAF mutations are susceptible to existing BRAF inhibitors, it is necessary to produce novel-generation BRAF inhibitors. The most prevalent BRAF mutation in lung cancer, particularly V600E, can lead to uncontrolled cell growth [313].

RTKs are characterized as key elements to go beyond BRAF inhibitor resistance and as appropriate targets for drugs, allowing for the more effective treatment of BRAF mutant tumors. RTKs are essential for the reactivation of MAPK/ERK signal transmission, the formation of bypassing signaling pathways, and the emergence of treatment resistance to BRAF inhibitors [320]. The dysregulation of RTKs, especially EGFR, HER2, and HER3, is implicated in resistance to BRAF inhibitors in BRAF mutant cancers. The inhibition of BRAFV00E often leads to the disruption of ERK-mediated negative feedback within the MAPK pathway, which is implicated in compensatory EGFR activation, the reactivation of downstream MAPK signaling, and therapeutic failure [321,322].

Therapeutic approaches with a combination of BRAF inhibitors and RTK inhibitors, such as EGFR inhibitors and pan ErbB inhibitors or monoclonal antibodies, have shown promising preclinical activity [320]. Additionally, the activation of other RTKs including MET/HGFR and AXL further contributes to adaptive resistance via alternative bypass pathways [323]. Novel dual inhibitors (targeting EGFR and RAF) and triple combinations (BRAF, MEK, and RTK inhibitors) have demonstrated promising efficacy in overcoming resistance. Therefore, RTKs represent both key mediators of resistance and promising therapeutic targets, which highlights their significance in refining treatment strategies for BRAF mutant cancers [320].

## 18. Conclusions

TK dysregulation results in cancer, unchecked cell division, and oncogenic conversion. Therefore, inhibiting TK can be a revolutionary advancement in cancer research, care, and treatment that clinically enhances quality of life. From this point of view, several preclinical studies are now being conducted to investigate TKIs. But, there are still a lot of unanswered questions regarding the mechanisms underlying TKIs and their selectivity. Numerous mechanisms, such as genetic mutations, changes in gene expression, and modifications to the tumor microenvironment, may result in TKI resistance. Additionally, the target gene may occasionally undergo alterations such as mutations and amplifications, allowing cells to proliferate even in the presence of an inhibitor. In such cases, strong kinase inhibitory activity from natural products looks very promising for lung cancer and, hence, NSCLC management. In this review, we discussed the fact that many plant-derived natural products including resveratrol, ginsenosides, astragalosides IV, BA, polyphyllin, and leelamines are reported to increase the anticancer activities of TKIs. However, these molecules are still not documented to inhibit TKs directly or indirectly. Rather, they function as pathway modulators or chemosensitizers affecting the potential of TKIs via complementary mechanisms. Therefore, their inclusion into TKI-based therapeutic strategies may provide a beneficial adjunct in lung cancer management. It is highly recommended to complete further research on the drug-like properties of such natural-product-derived molecules for the development of selective TKIs. Further, the research into bioactivity data and scaffolds obtained from natural sources for drug-screening important biological characteristics is necessary for the efficient treatment of lung cancer.

## Figures and Tables

**Figure 1 cimb-47-00498-f001:**
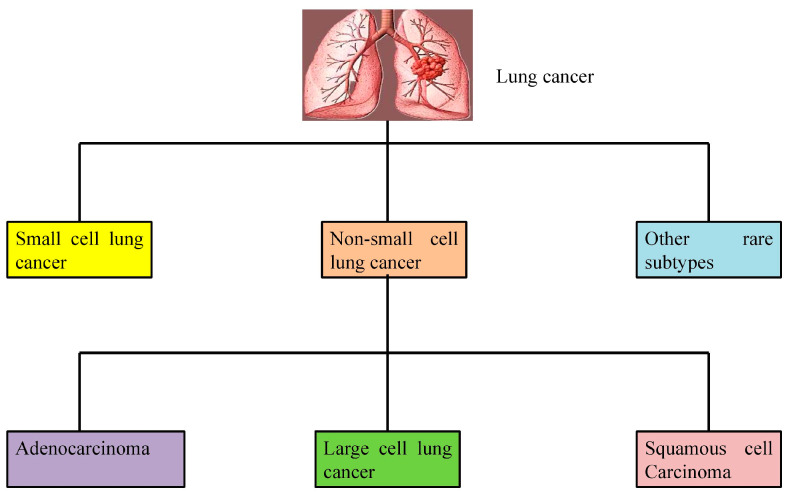
Types of lung cancer. Lung cancer is broadly classified into three categories including small cell lung cancer, non-small cell lung cancer, and other rare subtypes. Non-small lung cancer is further divided into squamous cell carcinoma, large cell lung carcinoma, and adenocarcinoma.

**Figure 2 cimb-47-00498-f002:**
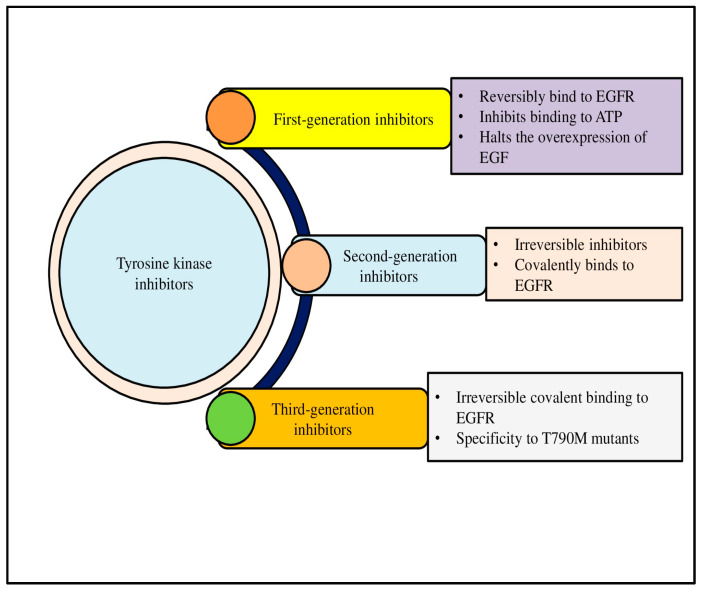
Tyrosine kinases (TKs) are crucial signaling proteins in cells that have a range of biological functions, such as promoting cell migration and proliferation. One systemic cancer therapeutic approach is the inhibition of angiogenic TKs. Based on generation, three kinds of antiangiogenic Tyrosine Kinase Inhibitors have been approved for use in patient therapy.

**Figure 3 cimb-47-00498-f003:**
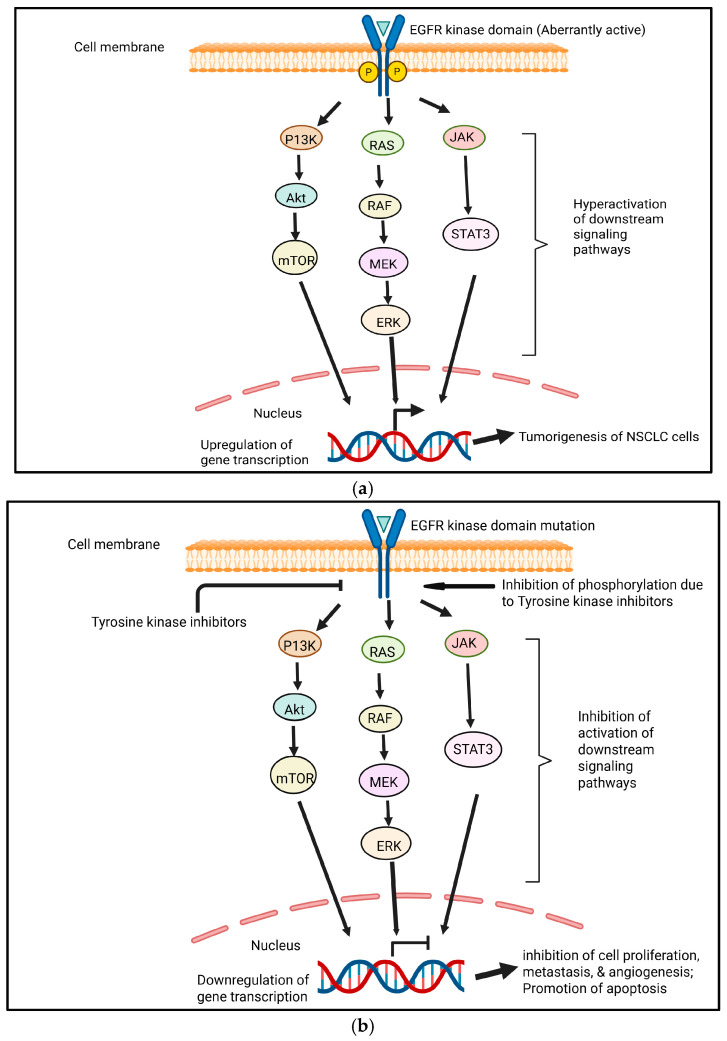
(**a**) EGFR pathways in NSCLC tumorigenesis. Aberrantly active EGFR leads to the continuous activation of signaling pathways, which results in the tumorigenesis of NSCLC cells. Created in https://BioRender.com. (**b**) The general inhibition of EGFR pathways due to tyrosine kinase inhibitors. Small-molecule tyrosine kinase inhibitors inhibit phosphorylation, resulting in the inhibition of the activation of downstream signaling pathways. This leads to the downregulation of genes involved in tumorigenesis. Cell proliferation, metastasis, and angiogenesis become inhibited. However, apoptosis becomes promoted. Created in https://BioRender.com.

**Table 1 cimb-47-00498-t001:** FDA-approved TKIs with their clinical trial numbers and outcomes of these trials are provided.

EGFR TKIs	Clinical Trial Number	Outcomes/Goals	References
Gefitinib, erlotinib, or icotinib	NCT02893332	For EGFR-mutated NSCLC, the addition of upfront local treatment utilizing RT statistically significantly improved PFS and OS as compared to a first-line TKI alone.	[62]
Erlotinib with gefitinib	NCT01024413	At the moment of peak response to EGFR-TKI, the amount of L858R dropped and hit its lowest level.	[63]
AZD9291	NCT01802632	Patients with EGFR T790M-mutant lung cancer who had seen disease progression after previous EGFR tyrosine kinase inhibitor therapy showed high levels of AZD9291 activity.	[64]
Osimertinib	NCT01802632	After EGFR-TKI treatment, osimertinib produced a high objective response rate, favorable progression-free survival, and persistent response in patients with EGFRm T790M advanced non-small cell lung cancer.	[65]
Brigatinib	NCT04318938	The primary goals of the ABP trial are to enhance knowledge of the underlying biology and enable the creation of a framework for the customized treatment of ALK+ NSCLC based on molecular characteristics, in addition to assisting enrolled patients in making treatment decisions.	[66]

**Table 2 cimb-47-00498-t002:** The efficacy of natural products such as tyrosine kinase inhibitors (TKIs) against a variety of malignancies has been shown in multiple studies, underscoring their promise for cancer treatment.

Natural Product	Type	Source	Cancer Type	Conclusion	Reference
Capsaicin	Alkaloids	*Capsicum annuum* L.	Human fibrosarcoma (HT-1080 cells)	Inhibits EGF-induced activation of MMP-9 and MMP-2, as well as tumor cell invasion and migration.	[90]
			Human fibrosarcoma cells	Capsaicin suppressed EGF-induced invasion and migration of human fibrosarcoma cells via EGFR-dependent FAK/Akt, PKC/Raf/ERK, p38 mitogen-activated protein kinase (MAPK), and AP-1 signaling, resulting in decreased matrix metalloproteinase 9 (MMP-9) expression.	[91]
Oxymatrine	Alkaloids	*Sophora* *flavescens* Aiton	Human malignant glioma (U251MG cells)	Reduced cell growth, halted the cell cycle at the G0/G1 phase, and reduced production of cell cycle regulatory proteins.	[92]
			Gastric cancer cells	Oxymatrine decreased the proliferation and invasion of gastric cells by inhibiting the EGFR/Cyclin D1/CDK4/6, EGFR/Akt, and MEK-1/ERK1/2/MMP2 pathways by inhibiting EGFRp-Tyr845.	[93]
Tetrandrine	Alkaloid	*Stephania tetrandra* S. Moore	Human colorectal adenocarcinoma (HT29 cells)	Inhibited the phosphorylation of EGFR and its downstream signaling pathways.	[94]
Apigenin	Flavone	Many fruits, vegetables, and seasonings	SKOV3 and SKOV3/TR cells	Apigenin’s cytotoxic action is explained by the considerable reduction in the mRNA and protein expression of Tyro3 and Axl receptor tyrosine kinases (RTKs).	[95]
			DLBCL cells	To promote G2/M phase arrest, apigenin inhibits cell cycle proteins such as CDK2/CDK4/CDK6/CDC2/p-RB. Our data mechanically show that apigenin inhibits the survival of DLBCL cells by significantly reducing the expression of the pro-proliferative pathway PI3K/mTOR.	[96]
Baicalein	Flavonoid	*Scutellaria baicalensis*, *Thymus vulgaris*, *Oroxylum indicum*	NSCLC	Induced apoptosis and changes in the regulation of cell cycle and altered expression of apoptotic regulatory proteins.	[97]
Curcumin	Flavonoid	*Curcuma longa* Lin	Breast cancer cell lines	By interfering with the p185neu protein’s binding to the chaperone GRP94, curcumin reduced the amount of p185neu in vivo and inhibited it in vitro.	[98]
			Non-small cell lung cancer (NSCLC) A549 and H460 cells	Curcumin has an anti-proliferative effect on both parental and chemo-resistant NSCLC cells via its new target, Axl RTK.	[99]
Luteolin	Flavone	Celery, carrots, peppers, thyme, oregano, etc.	NSCLC	Inhibits EGFR autophosphorylation.	[100]
Formononetin	Isoflavone	*Astragalus membranaceus*	NSCLC	Promoted the efficacy of EGFR-TKI by modulating the EGFR-AKT-Mcl-1 axis in a ubiquitination-dependent manner.	[101]
Fisetin	Flavonol	Apples, strawberries, cucumbers, persimmons, acacia plants and shrubs	Retinoblastoma angiogenesis	Blocked the VEGF/VEGFR signaling pathway.	[102]
Quercein	Flavonol	Fruits, vegetables, and beverages	Cisplatin-resistant ovarian cancer	An antiproliferative effect and the inhibition of lymphocyte TK activity were found.	[103]
Caffeic acid	Phenolic molecule	Coffee, wine, and tea, etc.	Breast cancer cells	Inhibits the phosphorylation of EGFR.	[104]
(-)-Epigallocatechin-3-gallate (EGCG)	Phenolics	*Camellia* *sinensis*	Hepatocellular carcinoma and colorectal cancer	Prevented the EGFR-TKs from activating.	[105]
			Breast cancer cell invasion	In MCF-7, MCF-7TAM, and MDA-MB-231 cells, EGCG and IIF treatments decreased the migratory tendency and changed the molecular network based on the interdependence of EGFR, CD44, and EMMPRIN expression.	[106]
			Colorectal adenomas	GTCs work as chemopreventive and anticancer agents by preventing the activation of certain RTKs, particularly EGFR, IGF-1R, and VEGFR2.	[107]
			NSCLC	Reduced the growth of erlotinib-sensitive and -resistant cell lines, including those with overexpressed c-Met and with developed erlotinib resistance.	[108]
			NSCLC	Treatment with EGCG decreases cell migration and alters the expression of vinculin and meta-vinculin.	[109]
			NSCLC	Combining EGCG derivatives with cisplatin inhibits the EGFR signaling pathway and reduces p-EGFR, p-AKT, and p-ERK expression both in vitro and in vivo.	[110]
Ellagic acid	Polyphenol	Fruits and nuts	Not specified	Dual inhibitor of VEGF and PDGF receptors indicated that it has significant antiangiogenic qualities.	[111]
Gossypol	Polyphenol	*Gossypium* sp.	Different cancer types in vitro and in vivo	Cell apoptosis but also autophagy, cell cycle arrest, and other abnormal cellular phenomena were found.	[112]
Honokiol	Phenolics	*Magnolia*	NSCLC	Reduced Akt phosphorylation and upregulated PTEN expression to downregulate the PI3K/Akt/mTOR pathway.	[113]
			Lung cancer	Regardless of the epidermal growth factor receptor (EGFR) mutation status, HNK specifically suppresses STAT3 phosphorylation, and STAT3 knockdown eliminated HNK’s anti-proliferative and anti-metastatic activities.	[114]
Magnolol	Hydroxylated biphenyl	*Magnolia* sps.	NSCLC	Multiple signaling pathways relayed by TKs were targeted.	[115]
Resveratrol	Stilbene	Grapes, peanuts, apples, blueberries, raspberries, etc.	HepG2 hepatocellular carcinoma cells	Effectively stops proliferation of cells, lowers reactive oxygen species generation, and triggers apoptosis, stopping the cell cycle in the G1 and G2/M phases. Additionally, it alters the NO/NOS system.	[116]
20(S)-ginsenoside Rg3	Triterpenoid saponins	*Panax ginseng*	Lung cancer	20(*S*)-Rg3 may inhibit CDK2, Cyclin A2, and Cyclin E1 by blocking the cell cycle at the G0/G1 phase.	[117]
Ginsenosides	Saponins	*Panax ginseng*	NSCLC	Decreased stemness marker expression and reduced spheroid formation were found.	[118]
			NSCLC	Inhibited the growth of NSCLC cells by activating the tumor suppressor p53-binding protein-1 and upregulating vaccinia-related kinase 1.	[119]
Astragaloside IV	Saponine	*Astragali* *Radix*	Lung cancer	AS-IV inhibited the M2 polarization of macrophages partially via the AMPK signaling pathway, which decreased the proliferation, invasion, migration, and angiogenesis of lung cancer.	[120]
			Breast cancer MDA-MB-231 cells	Orthotopic breast tumor development and lung metastases were inhibited by astragaloside IV.	[121]
Polyphyllin	Steroidal saponin	*Paris polyphylla*	NSCLC	Activation of SAPK/JNK pathway, suppression of p65 and DNMT1 expression.	[122]
			NSCLC	Downregulation of MALAT1 and inhibition of suppression of STAT3 phosphorylation.	[123]
Cucurbitacin E	Tetracyclic triterpenes	Cucurbitaceo-us plants	Non-small cell lung cancer (NSCLC) cell line A549	Cucurbitacins E exhibited anti-proliferative effect against A549 cells by targeting the EGFR/MAPK signaling pathway.	[124]
Cucurbitacin B			MKN-45 gastric carcinoma cells	The JAK2/STAT3 signaling pathway may be inhibited by cucurbitacin B, which decreases MKN-45 cell proliferation and promotes apoptosis.	[125]
Betulinic Acid	Triterpenoid	Many plants and herbs	Human NSCLC cell lines	In NSCLC cells, betulinic acid (BA) causes apoptosis and protective autophagy while suppressing cell division. BA’s ability to destroy cancer cells is increased by inhibiting autophagy.	[126]
Betulin	Triterpenoid	Many plants and herbs	Human chronic myelogenous leukemia cell line	Betulin modulates the mitogen-activated protein (MAP) kinase pathway with activity comparable to that of the well-known ABL1 kinase inhibitor imatinib mesylate.	[127]
Leelamine	Diterpene	Pine bark	Myelogenous leukemia cells	Modulation of the STAT5 pathway, autophagy, and death.	[128]

**Table 3 cimb-47-00498-t003:** Clinical phase trials of curcumin in lung cancer treatment.

ClinicalTrials.gov ID & Reference	Title of Study	Objective	Intervention/Treatment
NCT02321293 [159]	An Open-Label Prospective Cohort Trial of Curcumin Plus Tyrosine Kinase Inhibitors (TKI) for EGFR-Mutant Advanced NSCLC (CURCUMIN)	To evaluate the safety and tolerability of curcumin in combination with EGFR-TKIs in patients with advanced, non-resectable, and EGFR-mutant NSCLC	Dietary Supplement: CurcuVIVA™. Drug: Tyrosine Kinase Inhibitor Gefitinib (Iressa). Drug: Tyrosine Kinase Inhibitor Erlotinib (Tarceva).
NCT03598309 [160]	Phase II Trial to Modulate Intermediate Endpoint Biomarkers in Former and Current Smokers	To find out if an investigational combination drug called Lovaza (made with fish oils)+Curcumin C3 Complex (made from a root called curcumin) can help reduce the size of lung nodules	Drug: Curcumin C3 complex^®^. Drug: Lovaza^®^.
NCT04871412 [161]	The Thoracic Peri-Operative Integrative Surgical Care Evaluation Trial—Stage III (POISE)	Thoracic POISE project aims to improve outcomes by integrating complementary and individualized care approaches to enhance recovery, reduce adverse events, and extend survival in real-world clinical settings	Vit D, Provitalix Pure Whey Protein, Theracurmin 2X, Green Tea Extract, Trident SAP 66:33 Lemon, Probiotic Pro12.

**Table 4 cimb-47-00498-t004:** Combinational therapies of TKIs or chemotherapeutic drugs and natural products in lung cancer treatment.

Natural Product	Combined Drug (TKI/Chemotherapy)	Target	Observed Anticancer Effects	Reference
Emodin	Sorafenib, afatinib, cisplatin, paclitaxel, gemcitabine, and endoxifen	Lung adenocarcinoma and non-small cell lung cancer	Enhanced anticancer activity	[281]
*Scutellaria baicalensis*	Cisplatin	Lewis lung carcinoma cells, or LLC and in vivo C57BL/6J-tumor-inoculated mouse model	Showed synergistic effects, prevented tumor growth in vitro	[282]
*Marsdenia tenacissima*	Gefitinib	H460, H1975, and H292 cell lines	Caused apoptosis in resistant cells, decreased EGFR downstream signaling pathways	[283]
Sulforaphane	Gefitinib	Gefitinib-resistant lung cancer cells	Decreased the growth of gefitinib-resistant lung cancer cells, reversed gefitinib resistance, and inhibited the expression of SHH, SMO, and GLI1	[284]
Solamargine	Cisplatin	Lung cancer cell lines NCI-H1299 and NCI-H460	Increased apoptosis and anti-proliferative effects	[285]
Hederagenin	Paclitaxel, cisplatin	Lung cancer cells	Enhanced cytotoxicity via autophagy suppression	[286]
Narirutin	Cisplatin	Lung cancer cells	Dose-dependently inhibited TMEM16A, synergistic cytotoxicity	[287]
Curcumin	Cisplatin	NSCLC	Improved cisplatin efficacy	[288]
Curcumin	Cisplatin	Lung cancer cell lines A549, H460, and H1299	Improved cisplatin efficacy	[289]
*Phyllanthus emblica* and *Termanalia bellerica*	Doxorubicin, cisplatin	Lung and liver cells	Synergistic growth inhibition	[290]
5-Demethylnobiletin	Paclitaxel	Lung cancer cell lines	Synergistically inhibited proliferation	[291]
Aloe-emodin	Gefitinib	NSCLC	Increased sensitivity to gefitinib and reversed EMT	[292]
Chinese herbal medicine	Erlotinib, gefitinib, or icotinib	Advanced NSCLC	Increased PFS in patients with advanced NSCLC with an EGFR mutation	[293]
Yiqi Chutan Tang (YQCT)	Gefitinib	EFGR-TKI-resistant lung cancer cells	Reduced drug resistance and improved anticancer effects when associated with gefitinib, enhancement of apoptosis and autophagy	[294]
Methoxyflavanone derivative	Nilotinib, bosutinib, dasatinib, and ponatinib	A549 cells	Inhibited the proliferation of A549 cells in combination	[295]
Berberine	Osimertinib	MET-amplified osimertinib-resistant lung cancer	Reduced the survival of multiple MET-amplified EGFR-mutant osimertinib-resistant NSCLC cell lines and had increased apoptosis induction	[296]

## Data Availability

No datasets were generated or analyzed during the current study.

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
