# Peer review of "The Role of Plant-Derived Natural Products as a Regulator of the Tyrosine Kinase Pathway in the Management of Lung Cancer"

_cimb, 2025, doi:10.3390/cimb47070498_

Round 1
Reviewer 1 Report
Comments and Suggestions for Authors
There are many contents in this review, certainly interesting and interconnected. However, some parts are very repetitive, and the same concepts are sometimes reiterated across different sections. It should therefore be revised to make it more concise and easier to read. Considering both the content and the title, the review should also include more detailed information on the role that natural molecules play both in the interaction with EGFR and as adjuvants to EGFR-TKI therapies. Additionally, figures should be expanded and improved.
In 2025, there was a review on this topic in the European Journal of Medicinal Chemistry Reports, doi.org/10.1016/j.ejmcr.2025.100251. The information provided by the authors is somewhat different, but a greater focus on the subject could definitely make the review more interesting, as it is currently not very engaging due to the lack of detailed information and scarce references.
Below are my comments:
The sentence “Therefore, it is more recommended to produce novel medications using natural resources.” is repeated twice.
The sentence “Tyrosine kinase inhibitors (TKIs) decrease tyrosine kinase phosphorylation and compete with ATP for the ATP binding site of TK, which stops the growth of cancer cells [14].” is not entirely accurate because there are several types of TKIs. Although the authors focus on the ATP-binding pocket, it is important to provide readers with a comprehensive overview. In particular, TKIs can be ATP-competitive, allosteric, or bivalent (reviewed in doi.org/10.1021/acschembio.4c00028). The authors should mention the broad landscape of existing TKIs, including a review that explains the complexity and challenges in designing new TKIs.
Similarly, in line 272, when discussing first-, second-, and third-generation TKIs, it is important to also include information about fourth-generation TKIs, some of which are currently entering clinical trials.
A complex figure illustrating the role of the EGFR signaling pathway is needed.
Section 12: The table lists various natural molecules that have demonstrated antitumor activity by targeting different biological targets. For example, EGCG; the authors mention its ability to inhibit EGFR activation and state that EGCG exhibits inconsistent effects on the small number of NSCLC cell lines tested [212,213]. In reality, there are numerous studies on the action of EGCG on NSCLC, especially regarding EGFR, both wild-type and mutated, which makes it interesting as an adjuvant in cases of mutated EGFR. Some references include [DOI: 10.1002/ptr.7990; PMID: 19638461; DOI: 10.3390/ijms222111833; doi.org/10.1186/s12935-019-0981-0].
In section 13, the text is somewhat discursive, with concepts already discussed previously, adding little new information. The only new details are those in the last lines (lines 1666–1171). Additional examples from the literature should be discussed. Since this review focuses on the role of plant-derived natural products as regulators of the tyrosine kinase pathway in lung cancer management, the section on management should be expanded.
A figure illustrating the chemical structures of the molecules discussed in the text is necessary.
Author Response
There are many contents in this review, certainly interesting and interconnected. However, some parts are very repetitive, and the same concepts are sometimes reiterated across different sections. It should therefore be revised to make it more concise and easier to read. Considering both the content and the title, the review should also include more detailed information on the role that natural molecules play both in the interaction with EGFR and as adjuvants to EGFR-TKI therapies. Additionally, figures should be expanded and improved.
- In 2025, there was a review on this topic in the European Journal of Medicinal Chemistry Reports, doi.org/10.1016/j.ejmcr.2025.100251. The information provided by the authors is somewhat different, but a greater focus on the subject could definitely make the review more interesting, as it is currently not very engaging due to the lack of detailed information and scarce references.
Response- We sincerely thank the reviewer for bringing this recent publication to our attention. We have carefully reviewed the article and acknowledge its relevance to our topic. We have accordingly revised our manuscript and cited this and highlighted in the revised version.
- Below are my comments:
- The sentence “Therefore, it is more recommended to produce novel medications using natural resources.” is repeated twice.
Response- Thank you for pointing this out, the repeated sentence has been removed to avoid redundancy.
- The sentence “Tyrosine kinase inhibitors (TKIs) decrease tyrosine kinase phosphorylation and compete with ATP for the ATP binding site of TK, which stops the growth of cancer cells [14].” is not entirely accurate because there are several types of TKIs. Although the authors focus on the ATP-binding pocket, it is important to provide readers with a comprehensive overview. In particular, TKIs can be ATP-competitive, allosteric, or bivalent (reviewed in doi.org/10.1021/acschembio.4c00028).
Response- Thank you for this insightful comment. We have revised the manuscript and added a sentence to accurately reflect the nature of TKIs, including ATP-competitive, allosteric, or bivalent nature. We cited the recommended article.
2.3. The authors should mention the broad landscape of existing TKIs, including a review that explains the complexity and challenges in designing new TKIs.
Response- Thank you for your valuable suggestion, we have added a new section (Section 11) with title “Expanding therapeutic landscapes of kinases: Complexities and challenges in de-signing novel TKIs”, to provide a more comprehensive and updated perspective on TKIs. To avoid redundancy, we ensured that the content does not repeat earlier sections. We believe this addition complements the existing content of this manuscript and enhances the relevance and depth of this manuscript.
- Similarly, in line 272, when discussing first-, second-, and third-generation TKIs, it is important to also include information about fourth-generation TKIs, some of which are currently entering clinical trials.
Response- Thank you for this insightful comment, we have revised the manuscript by adding a new section (Section 15) with title “Fourth generation TKIs.”
- A complex figure illustrating the role of the EGFR signaling pathway is needed.
Response- We have modified figure 3a, which illustrates the role of EGFR signaling pathway in NSCLC tumorigenesis.
- Section 12: The table lists various natural molecules that have demonstrated antitumor activity by targeting different biological targets. For example, EGCG; the authors mention its ability to inhibit EGFR activation and state that EGCG exhibits inconsistent effects on the small number of NSCLC cell lines tested [212,213]. In reality, there are numerous studies on the action of EGCG on NSCLC, especially regarding EGFR, both wild-type and mutated, which makes it interesting as an adjuvant in cases of mutated EGFR. Some references include [DOI: 10.1002/ptr.7990; PMID: 19638461; DOI: 10.3390/ijms222111833; doi.org/10.1186/s12935-019-0981-0].
Response- Thank you for your valuable suggestion, we have revised the table 2 by adding the suggested content and added recommended articles.
Note- After adding new section according to suggestions of our respected reviewers, section 12 became section 13.
- In section 13, the text is somewhat discursive, with concepts already discussed previously, adding little new information. The only new details are those in the last lines (lines 1666–1171). Additional examples from the literature should be discussed. Since this review focuses on the role of plant-derived natural products as regulators of the tyrosine kinase pathway in lung cancer management, the section on management should be expanded.
Response- We sincerely thank to our esteemed reviewers for their insightful comments. We have reduced the repetitive count and added new content related to combinational therapy.
Note- After adding new section according to suggestions of our respected reviewers, section 13 became section 14.
2.7. A figure illustrating the chemical structures of the molecules discussed in the text is necessary.
Response- We sincerely thank to our esteemed reviewers for their valuable suggestion regarding the inclusion of a figure illustrating the chemical structures of the molecules discussed. We fully agree that such visual presentation will increase the clarity and accessibility of the manuscript. However, considering already extensive length and comprehensive nature of the current review, incorporating an additional figure with numerous chemical structures may further significantly increase the size of manuscript. Further, we have a very short time for revision for lengthy comments from other esteemed reviewers; therefore we are not able to insert these chemical structures in this review. The chemical structures of these compounds are already available online.
Reviewer 2 Report
Comments and Suggestions for Authors
[CIMB] Manuscript ID: cimb-3700020
The manuscript by Faris Al-rumaihi et al. is very interesting and well-written. The authors describe the role of Tyrosine Kinases in the process of carcinogenesis, focusing on the molecular pathways and molecular targets that lead to the development of lung cancer. Moreover, the authors report the anticancer and pro-apoptotic effects of many phytochemicals: curcumin, resveratrol, ginsenosides, astragaloside IV, cucurbitacin, apigenin, quercetin, betulinic acid, EGCG, polyphyllin, gossypol, formononetin, leelamine, ellagic acid, luteolin, fisetin, baicalein, caffeic acid. The most interesting chapters describe the improved anticancer effects of the combination of Tyrosine Kinase Inhibitors (TKI) drugs and natural molecules to inhibit the molecular mechanisms of drug resistance of lung cancer cells, decrease the pro-survival mechanisms of tumor cells, and increase the levels of the pro-apoptotic markers, such as caspase 3. The authors should answer my major and minor requests to improve their Manuscript.
MAJOR REVISION
1)The authors should add in Chapter 13 of their Manuscript a new Table, which summarizes the modulated molecular targets and the anticancer effects exerted by the combination of TKI drugs and the natural molecules described in Chapter 12 of their Manuscript.
MINOR REVISIONS
1)The authors should describe the activation of the initiator caspases 8 and 9 induced by the phytochemicals reported in their manuscript in both in vitro and in vivo experimental models of lung cancer.
2)The authors should improve Figure 3b because it is nearly identical to Figure 3 a. In fact, a clear graphic representation of the TKI-mediated inhibition of both abnormal growth of cancer cells and of tyrosine kinase receptors should be added to the Figure.
Author Response
The manuscript by Faris Al-rumaihi et al. is very interesting and well-written. The authors describe the role of Tyrosine Kinases in the process of carcinogenesis, focusing on the molecular pathways and molecular targets that lead to the development of lung cancer. Moreover, the authors report the anticancer and pro-apoptotic effects of many phytochemicals: curcumin, resveratrol, ginsenosides, astragaloside IV, cucurbitacin, apigenin, quercetin, betulinic acid, EGCG, polyphyllin, gossypol, formononetin, leelamine, ellagic acid, luteolin, fisetin, baicalein, caffeic acid. The most interesting chapters describe the improved anticancer effects of the combination of Tyrosine Kinase Inhibitors (TKI) drugs and natural molecules to inhibit the molecular mechanisms of drug resistance of lung cancer cells, decrease the pro-survival mechanisms of tumor cells, and increase the levels of the pro-apoptotic markers, such as caspase 3. The authors should answer my major and minor requests to improve their Manuscript.
- MAJOR REVISION
- The authors should add in Chapter 13 of their Manuscript a new Table, which summarizes the modulated molecular targets and the anticancer effects exerted by the combination of TKI drugs and the natural molecules described in Chapter 12 of their Manuscript.
Response- We sincerely thank the reviewer for this excellent suggestion. In response, we have edited the section according to our another esteemed reviewer and added a new table (table 4) in this section 13 summarizing the modulated molecular targets and anticancer effects associated with the combination of TKIs/chemotherapeutic drugs and the natural compounds according to your valuable suggestions. While we aimed to align this table with the compounds discussed in Section 12, we found limited direct evidence supporting combination therapies for some of those specific natural products. Therefore, we have curated combinations based on the most relevant and well-documented studies to provide a comprehensive and scientifically grounded overview. We believe this addition significantly enhances the translational value of the manuscript.
Note- As we have added a new section according to the suggestions our another highly respected reviewer, your suggested section is section 14 in the revised manuscript.
MINOR REVISIONS
- The authors should describe the activation of the initiator caspases 8 and 9 induced by the phytochemicals reported in their manuscript in both in vitro and in vivo experimental models of lung cancer.
Response- Thanks for your valuable suggestions, we have added a new section (section 16) with a title “.Caspase-mediated apoptosis induced by phytochemicals in lung cancer models.”
2) The authors should improve Figure 3b because it is nearly identical to Figure 3 a. In fact, a clear graphic representation of the TKI-mediated inhibition of both abnormal growth of cancer cells and of tyrosine kinase receptors should be added to the Figure.
Response- We sincerely thank the reviewer for pointing out the mistake. We have revised the figure 3a and 3b both.
Reviewer 3 Report
Comments and Suggestions for Authors
This is a comprehensive review of the potential of natural products as TK-targeting compounds in lung cancer diseases. The content is meaningful, relevant research is cited mostly. However, there are some flaws in the manuscript and I recommend major revision for the following reasons:
Please write scientific species names in italics throughout the manuscript.
Line 283: Please correct ´´haepatocellular´´. Line 493: Please correct ´´approva´´.
Figure 1: The hyphen is missing in ´´Non small´´. Please correct
Sections 9 and 10: Please discuss briefly the role of RTKs and its inhibitors for the development and treatment of BRAF-mutant lung cancers and cite pertinent research and/or review literature dealing with this matter.
Figure 3: The arrow behind ´´Angiogenesis´´ collides with the s letter. Please modify.
Section 12: The authors mention honokiol and capsaicin in Table 2 but they are not mentioned in the main text. Vice versa, some natural products described in the main text were not mentioned in the table. Please adjust the manuscript accordingly.
Section 12: What about the close honokiol analog magnolol? Does magnolol also show relevant effects?
Section 12: The reason behind the order of the natural products described in the main text is unclear. Maybe the authors can rearrange and combine compound classes such as flavonoids.
12.1.: Are there clinical studies with curcumin in lung cancer patients? Please discuss.
Table 2: Correct ´´Sinensis´´ (small s as first letter). Please adjust the text in the lines properly (e.g., Oxymatrine – Alkaloids and Betulinic acid – Triterpenoid are in different lines in my copy of the manuscript). Add ´´Betulin´´ to the natural product column (in addition to Betulinic Acid) since it is mentioned in the conclusion column of this table.
References: The references are inconsistent in style. Title words were written with big starting letters or not. Some journal names were abbreviated others not. Please provide a consistent references list, best in the style recommended by the journal.
Author Response
- Please write scientific species names in italics throughout the manuscript.
Response- We are highly thankful to our esteemed reviewer. We have italicized scientific names throughout the manuscript.
- Line 283: Please correct ´´haepatocellular´´. Line 493: Please correct ´´approva´´.
Response- We sincerely thank the reviewer for pointing out the mistake. we have revised and highlighted them
- Figure 1: The hyphen is missing in ´´Non small´´. Please correct.
Response- We sincerely thank the reviewer for pointing out the mistake. We have included the hyphen.
- Sections 9 and 10: Please discuss briefly the role of RTKs and its inhibitors for the development and treatment of BRAF-mutant lung cancers and cite pertinent research and/or review literature dealing with this matter.
Response- We are highly thankful for the reviewers for their valuable suggestions. We added a new section (section 17) with a title “Role of RTKs and its inhibitors for the development and treatment of BRAF-mutant lung cancers.”
- Figure 3: The arrow behind ´´Angiogenesis´´ collides with the s letter. Please modify.
Response- We sincerely thank the reviewer for pointing out the mistake. We redraw the figures 3a and 3b.
- Section 12: The authors mention honokiol and capsaicin in Table 2 but they are not mentioned in the main text. Vice versa, some natural products described in the main text were not mentioned in the table. Please adjust the manuscript accordingly.
Response- We are very thankful to our respected reviewer for such valuable suggestion of reviewer. We adjusted the table and text according the valuable suggestions of our esteemed reviewer.
After revising the manuscript, section 12 became section 13, please check this section.
- Section 12: What about the close honokiol analog magnolol? Does magnolol also show relevant effects?
Response- We are very thankful to our respected reviewer for such valuable suggestion of reviewer. We added a paragraph about magnolol in section 13 and highlighted it.
- Section 12: The reason behind the order of the natural products described in the main text is unclear. Maybe the authors can rearrange and combine compound classes such as flavonoids.
Response- Thanks for your valuable suggestion. We combined the natural products according to their classes.
12.1.: Are there clinical studies with curcumin in lung cancer patients? Please discuss.
Response. Thanks for a valuable comment. We have provided a table (table 3) summarizing the clinical studies for the implication of curcumin in lung cancer treatment and highlighted.
- Table 2: Correct ´´Sinensis´´ (small s as first letter). Please adjust the text in the lines properly (e.g., Oxymatrine – Alkaloids and Betulinic acid – Triterpenoid are in different lines in my copy of the manuscript). Add ´´Betulin´´ to the natural product column (in addition to Betulinic Acid) since it is mentioned in the conclusion column of this table.
Response- We are sincerely thankful to our respected reviewer for figuring out the mistake. We corrected the word and highlighted it. We added betulin in the main text and adjusted the table and text by balancing the content in both according to the valuable suggestions of our respected reviewer.
- References: The references are inconsistent in style. Title words were written with big starting letters or not. Some journal names were abbreviated others not. Please provide a consistent references list, best in the style recommended by the journal.
Response- We are sincerely thankful to our reviewer for this meaningful comment. The MDPI has mentioned that they will make a consistent list of reference by themselves in the proof read file. We apologize to our respected reviewer.
Round 2
Reviewer 1 Report
Comments and Suggestions for Authors
The authors made most of the required changes and the review is now suitable for publication.
Reviewer 2 Report
Comments and Suggestions for Authors
The authors answered all my Major and Minor requests and, in my opinion, the revised version of their Manuscript can now be accepted for publication in CIMB Journal.
Reviewer 3 Report
Comments and Suggestions for Authors
The authors have improved the manuscript in accordance with my comments and the revised manuscript is suitable for publication now.